# Quantifying phage-host dynamics using droplet microfluidics

Louis Givelet ⬢ , Sophie von Schönberg ⬢ , Florian Katzmeier ⬢ &
Friedrich C. Simmel ⬢ ✉

Since their discovery, bacteriophages—viruses that infect bacteria—have been invaluable to molecular biology and biotechnology. Renewed interest in phage-based antimicrobials, driven by the global antibiotic resistance crisis, highlights the need for improved quantitative tools. While conventional double-layer plaque assays (DLA) have provided foundational insights, they are limited by their inability to monitor infection dynamics over time and the inflexibility in experimental setups. Here, we present a high-throughput droplet microfluidics platform to quantify individual phage infection events. By co-encapsulating individual phages and bacteria in microfluidic droplets, we precisely control key experimental parameters such as exposure time and the ratio of phages to bacteria. This approach enables direct quantification of lysis events and measurement of lysis kinetics without interference from further progeny-driven infection processes inherent to bulk cultures. Applicable to diverse phage-host systems, this method offers a dynamic and accurate framework for studying phage biology and supports the development of phage-based antimicrobial strategies.

Since the first discoveries of bacteriophages more than a century ago[1,2], their longstanding investigation has had a major impact on biological research, both at the fundamental level and in terms of applications. Being recognized as the most abundant biological entity on earth[3], phages are ubiquitously present in extremely diverse environments[4], typically outnumbering their bacterial hosts by an order of magnitude. Phages are pervasive to prokaryotic life and therefore exert significant selective pressure on microbial communities. Through co-evolution with their hosts, they increase the rate of molecular evolution in bacteria and play a pivotal role in horizontal gene transfer, thus substantially shaping the complexity of ecosystems[5–7].

Bacteriophages, as obligate parasites, exist in a metabolically inactive virion form outside of their host cells, entirely dependent on infecting bacterial hosts for replication. This fundamental simplicity has been instrumental in elucidating several key concepts in biology, and thus had a major impact in establishing the foundational principles of molecular biology[8]. Bacteriophage research has resulted in a multitude of applications in molecular biology and biotechnology, such as type II restriction enzymes[9], DNA ligases[10], phage display[11] or CRISPR-Cas tools[12].

Besides the immense biotechnological potential of phage biology, an emerging health crisis has renewed the interest in phage studies in the scientific community in recent years: the rise of antibiotic resistance has become a pressing public health concern, prompting warnings of the onset of a post-antibiotic era[13–15]. Utilizing bacteriophages as therapeutic agents in combating bacterial infections is increasingly considered as a promising alternative to antibiotics[16–18]. Notably, this potential was already recognized shortly after the initial discovery of phages[2,19], but could not be fully exploited so far.

One of the most robust methods to study phage-host interactions is the double layer plaque assay (DLA), used primarily for quantification of phages[20,21]. In this technique, host cells and phages are incubated inside a thin layer of soft nutrient agar (usually atop a bottom layer of nutrient agar), allowing diffusion of phages as well as cell growth inside the agar layer. Deposited phages infect bacteria, replicate within them and are released after bacterial lysis. Repeated cycles of infection, replication and lysis result in the creation of plaques: circular lysis zones, visible as clear areas that are straightforward to count. The technique thus allows quantifying the virulence of a phage

Department of Bioscience, TUM School of Natural Sciences, Technical University Munich, Am Coulombwall 4a, Munich, Germany. ✉e-mail: simmel@tum.de

solution with respect to a specific bacterial host, measured in terms of Plaque Forming Units/mL (PFU/mL).

Even though DLA is considered the gold standard for phage quantification since its invention in 1936[20,22], it suffers from a couple of significant drawbacks. DLA represents a fixed and rather inflexible experimental setup and can suffer from poor reproducibility[23,24]. Successful titer determination is dependent on certain properties of both host and phage: the ability of the host to grow to sufficient density, as well as the release of an adequate number of progeny phages to form a discernible plaque. The dependence on host growth rate also dictates that - for the majority of bacterial hosts - DLA requires at least overnight incubation, for slow-growing strains it may even require incubation for several days[20,24]. Additionally, the semi-solid agar environment not necessarily represents a physiological environment for all phage-host pairs. Traditional alternatives for the enumeration of phages are microscopy-based methods such as TEM or molecular methods like qPCR[20]. Lastly, DLA is not suited for kinetic studies of phage infection[24]. Indeed, the first time point where plaques can be identified is reached only when the bacterial host has grown to sufficient density and is close to stationary phase. As a consequence, the dynamics of phage infection are usually not monitored over time using plaque assays, but rather in bulk liquid culture measuring optical density[25] or using fluorescent DNA dyes[24]. However, all of these methods characterize phage infection at the population level, where the continuous release of newly formed phages prevents the study of individual phage infection events.

In the present work, we develop a novel, statistics-guided approach for quantifying individual phage infection events using high-throughput droplet microfluidics. Over the past decade, droplet microfluidics has become an important enabling technology for diverse bioanalytical applications[26–30]. It has been employed to isolate and amplify phages[31–34], and to develop various droplet-based methods for phage detection and enumeration, relying on digital PCR[35], heterologous protein expression[36], DNA intercalating dyes[33,37], bacterial growth rate monitoring[38] or light scattering[39]. With the exception of recent work by Nikolic et al.[40], however, the potential of droplet microfluidics for investigating infection dynamics at the single-droplet level has not been fully realized.

Here, we introduce several conceptual and technical advances that distinguish our approach from existing methods. First, we employ a dual co-flow encapsulation scheme, in which phage- and bacteria-containing solutions are introduced separately and meet only at the moment of encapsulation. This ensures that phage exposure occurs precisely and synchronously, eliminating pre-encapsulation interactions that would otherwise bias infection kinetics or titer quantification. Second, by systematically varying droplet sizes and mixing fractions within the same emulsion, we can explore how infection outcomes depend on compartment volume and phage-to-bacteria ratios. Varying the mixing fraction can be used to expand the dynamic range and allows the phage titer to be determined with greater confidence. Third, we can provide robust statistics, as our optical detection system routinely analyzes up to $10^5$ droplets per experiment, acquiring thousands of events per second without the need for image-based analysis. Finally, our theoretical framework explicitly accounts for the encapsulation statistics of both host and phage, allowing operation across a wide range of bacterial densities and enabling kinetic modeling of lysis dynamics in the droplets.

Together, these features establish a quantitative and generalizable platform for studying bacteriophage infection dynamics. In contrast to bulk measurements, which provide only averaged population-level information, our approach allows the controlled and parallelized investigation of thousands of independent phage-host encounters, each defined by known multiplicities and compartment volumes. Moreover, it directly detects lysis events rather than free DNA templates such as in digital PCR, thereby providing an absolute and infection-relevant quantification of phage activity.

## Results

### A high-throughput digital phage assay

In previous work that utilized droplet microfluidics for counting or analysis of bacteriophages, the phages were invariably mixed with their bacterial hosts in a single bulk suspension before they were compartmentalized into droplets for further study. While this approach is straightforward, it requires the rapid encapsulation of phages and bacteria after mixing, in particular when studying phages with a short lysis cycle such as T7 phage[41]. In our workflow, suspensions containing phages and bacteria are co-injected into a flow-focusing microfluidic chip to ensure that the exposure of the bacteria to the phages begins precisely at the moment of encapsulation (Fig. 1a). For the detection of bacterial lysis events in the droplets, our method relies on the labeling of the bacterial DNA released after cell lysis by the cell wall-impermeable intercalating fluorescent DNA dye YOYO-1 (Fig. 1b). Even though YOYO-1 is also capable of labeling phage DNA[33,42], we expected droplets containing lysed hosts to display substantially higher fluorescence levels than droplets containing only phages due to the sheer size difference between the phage and bacterial genomes. We confirmed that this is indeed the case using bulk assays (see Figure S1). After collection and incubation of the droplet emulsions for a defined time interval, the emulsions are re-injected into a separate microfluidic screening chip and scanned at rates of up to several kHz using a dedicated high-speed fluorescence detection setup (Fig. 1c, Figure S2). For each droplet, a red and green fluorescence emission signal is acquired. During production of the droplet emulsions, we introduce a red fluorescent reference dye (Atto 565), which allows to collect additional information on the droplets such as their size and the mixing fraction $\alpha$ between the phages and the bacteria. $\alpha$ is defined as the volume fraction of the phage suspension in the droplet, and thus the bacterial suspension has a mixing fraction of $1 - \alpha$. Based on the green fluorescence signal distribution caused by YOYO-1, droplets are digitized (defined as positive for lysed and negative for non-lysed or 1 and 0, respectively) to determine the digital titer (Fig. 1d).

### Statistics of bacterial lysis in droplets

To obtain the phage titer $c_p$ of the encapsulated solution, we have to relate it to the fraction of positive droplets that exhibit a lysis event and thus become fluorescent. This depends on the statistics of the number of encapsulated phages and bacteria. To derive the probability $P_L$ that a droplet exhibits a lysis event, we assume that every encapsulated infective phage will lead to the lysis of a bacterium when one is available. In other words, for a droplet to exhibit a lysis event, it must encapsulate at least one bacterium and at least one phage. Here, we only consider phages that will infect and lyse host cells under the experimental conditions, as in the traditional DLA titration method. Assuming independent encapsulation of each phage and bacterium, we can model the probabilities $P_b$ and $P_p$ of encapsulating exactly $b$ bacteria and $p$ phages within a droplet using Poisson distributions[43]. These distributions are each characterized by a single parameter, the expected numbers $\lambda_b$ and $\lambda_p$ of encapsulated bacteria and phages, respectively. $\lambda_p$ can be calculated from the phage stock titer $c_p$, the droplet volume $V$, and the mixing fraction $\alpha$ as $\lambda_p = \alpha c_p V$. Similarly, the expected number of encapsulated bacteria is given by $\lambda_b = (1 - \alpha)c_b V$, where $c_b$ is the bacterial stock density. According to the Poisson distribution, the probability of having exactly zero phages encapsulated is $P_{p=0} = \exp(-\lambda_p)$, and thus the probability of having at least one phage encapsulated is given by $P_{p>0} = 1 - \exp(-\lambda_p)$. Similarly, the probability of having at least one bacterium encapsulated in the droplet is $P_{b>0} = 1 - \exp(-\lambda_b)$. Thus, the probability $P_L$ that a droplet contains at

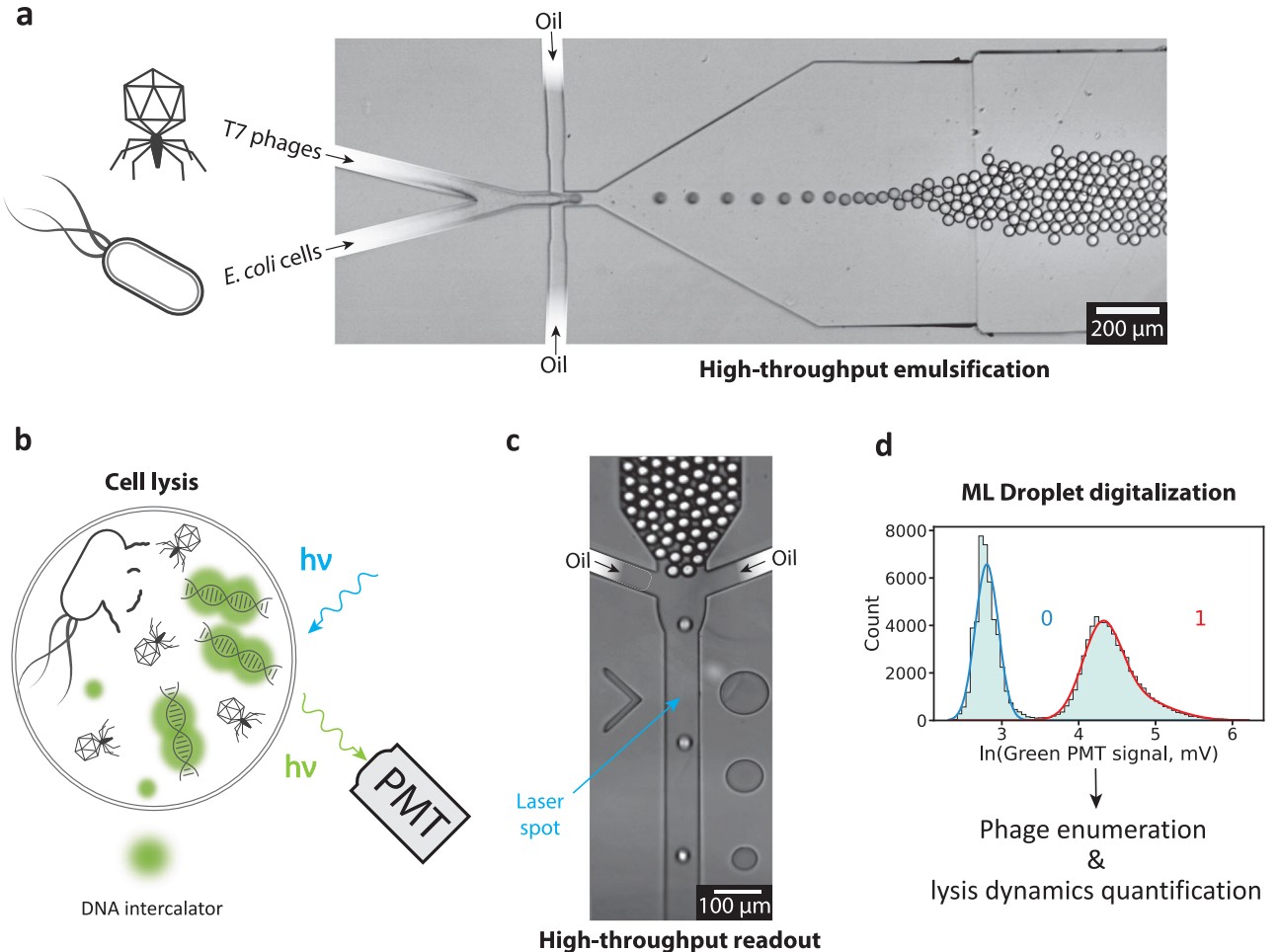

**High-throughput emulsification**

**Cell lysis**

**High-throughput readout**

**ML Droplet digitalization**

Phage enumeration
&
lysis dynamics quantification

**Fig. 1 | Experimental workflow. a** Bright-field microscopy image showing the co-encapsulation of phages and bacteria. Emulsions are incubated at 37°C before acquisition. This protocol of emulsification was repeated in 20 independent experiments. **b** Close-up schematic of lysis in a droplet. A DNA intercalator (YOYO-1) leads to increased green fluorescence emission upon lysis. PMT: Photo-Multiplier Tube. **c** Bright-field microscopy image of high-throughput droplet acquisition. The spacing oil ensures that the droplets are well separated and excited by the laser sequentially. This protocol of acquisition was repeated in 20 independent experiments. **d** The green fluorescence signal allows digitizing the droplets using a Gaussian Mixture Model (GMM) to compute the digital phage titer and other parameters. The computation takes into account various factors, such as the mixing ratio between phage and bacterial suspensions, droplet volume, incubation time, and bacterial cell density.

least one bacterium and at least one phage is given by

$$P_L = P_{b>0} \cdot P_{p>0} = \left(1 - e^{-\lambda_p}\right)\left(1 - e^{-\lambda_b}\right). \quad (1)$$

Based on our assumption, $P_L$ corresponds to the fraction of droplets that are identified as positive, which is evidenced by their higher fluorescence signal. Equation (1) thus allows the calculation of the phage titer $c_p$ for a given droplet volume $V$, bacteria density $c_b$, and mixing fraction $\alpha$ from $P_L$, which can be experimentally determined through the observation of a large enough number of droplets. Note that for large enough $\lambda_b$, $P_L \approx 1 - e^{-\lambda_p}$ becomes independent of the bacterial density.

**Dynamic range of the digital titer**

To test the functionality and performance of our system, we measured the green fluorescence levels of droplets for various compositions (Fig. 2). When phages and bacteria were co-encapsulated, we observed a multimodal distribution for the green fluorescence signal (Fig. 2a & c), whereas distributions for droplets containing only bacteria (Fig. 2e) or only reference dye (Fig. 2f) were monomodal. The fluorescence distribution of bacteria-only droplets displays a slight tail towards higher fluorescence values, which is absent in droplets containing only

reference dye. We attribute this minor increase to uncontrolled release of bacterial DNA, which is the result of unspecific cell death occurring during bacterial culture and preparation. During digitization, droplets belonging to the lower fluorescence mode are categorized as negative (0), whereas droplets displaying a high level of fluorescence are classified as positive (1).

Interestingly, encapsulating the suspension with a high nominal phage titer resulted in a multimodal distribution of the fluorescence signals for the positive droplets (Fig. 2a). By contrast, droplets formed with a low-titer suspension resulted in a monomodal positive distribution (Fig. 2c). This suggests that the higher positive fluorescence mode visible in Fig. 2a represents droplets in which several lytic cycles took place, resulting in larger amounts of free DNA in solution, a high phage count, and the lysis of multiple bacteria.

To achieve droplet digitization, we rely on a Gaussian Mixture Model (GMM), which enables automated separation of the droplet subpopulations. This approach eliminates the need for a fixed fluorescence threshold, as the GMM probabilistically assigns droplets to positive or negative classes even when their distributions overlap. It also minimizes the requirement for user input and helps to avoid user bias in this critical step. The droplet fluorescence distributions can be well described by log-normal distributions, and we therefore apply the

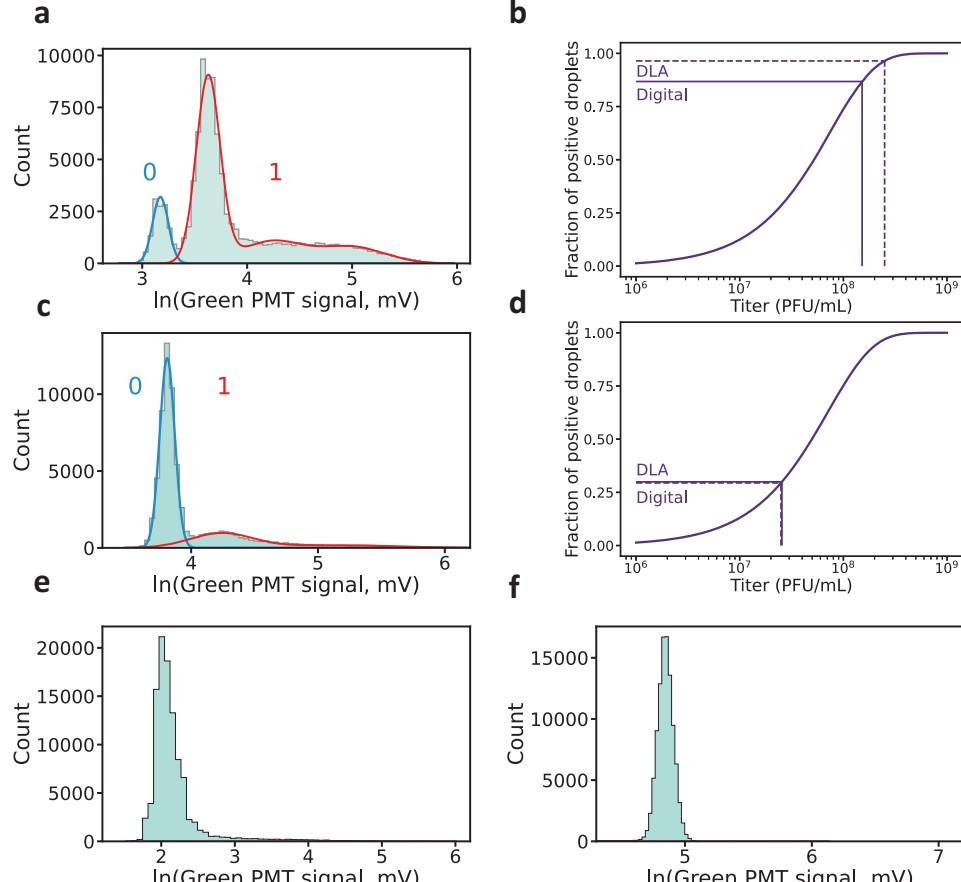

**Fig. 2 | Droplet digitization and digital titer determination.** Panels **a** and **c** show the logarithm-transformed green fluorescence signal distribution (light blue bars) for droplets containing bacteria and phages with (**a**) high and (**b**) low DLA phage titer, respectively. Modes detected by a Gaussian mixture model are represented by blue (0, negative droplets) and red curves (1, positive droplets). $P_L$ is determined by computing the sum of all of the higher mode contributions normalized by the total number of counts (cf. SI). **b**, **d** show the theoretical relationship between phage titer and observed fraction of positive droplets $P_L$. Vertical dotted lines indicate the titer determined by DLA, and horizontal dotted lines indicate the expected $P_L$ based on

DLA titer. Conversely, digital titers (vertical solid lines) are calculated from the experimentally determined $P_L$ (horizontal solid line). Their computed digital titers of (**b**) $1.516 \times 10^8$ PFU/mL ($n= 1.01 \times 10^5$ droplets) and (**d**) $2.553 \times 10^7$ PFU/mL ($n= 6.69 \times 10^4$ droplets) lie close to the titers determined through DLA, $2.50 \times 10^8$ PFU/mL and $2.50 \times 10^7$ PFU/mL, respectively. **e** Logarithm-transformed green fluorescence signal distribution without phages, $n=1.031 \times 10^5$ droplets. **f** Logarithm-transformed green fluorescence signal distribution of droplets containing only red reference dye, n= $1.001 \times 10^5$ droplets. For the acquisition of this blank measurement, the PMT gain was increased compared to the other panels.

GMM to the logarithm-transformed green fluorescence signals. Through the GMM, one can determine the fraction of positive droplets $P_L$. Equation (1) then allows to determine the parameter $\lambda_p$, which can be converted into a digital phage titer via $c_p = \lambda_p/(\alpha V)$. Digital titers obtained in this manner are calculated without prior knowledge of the DLA titer, but nonetheless exhibit close agreement with these values.

As can be observed in Fig. 2b & d, the relation between $P_L$ and phage titer flattens out for very low and very large titers, suggesting an optimal intermediate range, where the two values are well correlated. Outside of this range, small changes in the fraction of positive droplets translate to large variations in the computed digital titer, making its determination unreliable. Notably, as our system allows the analysis of large numbers of droplets ($10^4$-$10^6$), even extremely unbalanced proportions between positive and negative droplets can be theoretically determined with strong statistical confidence. Nevertheless, digital titers lying outside this dynamic range will inevitably suffer from a higher imprecision. This is especially true for small titers, where unspecific bacterial lysis, as observed in Fig. 2e, can lead to false positives.

We note that if an experimenter encounters a complex signal distribution, a convenient approach is to vary the mixing fraction. Observing how the different modes evolve with changes in the mixing fraction can help clarify their identity and prevent misinterpretation.

## Screening with discrete mixing fractions

Varying the mixing fractions $\alpha$ within the droplets also provides a way to expand the dynamic range of the digital titer. To this end, we dynamically changed the pressures at the two inlets of the aqueous solution while maintaining a constant overall dispersed phase pressure to ensure a uniform droplet size. Subpopulations of droplets with five different $\alpha$ values were generated and then identified via the fluorescence level of the red reference dye as described above (Fig. 3a). The numerical values for $\alpha$ were determined by analyzing the corresponding microscope images (Fig. 3b).

A combined analysis of the subpopulations theoretically results in an expansion of the dynamic range. Smaller mixing fractions $\alpha$ are more appropriate for determining high titers, whereas larger mixing fractions $\alpha$ are better suited for estimating low titer values (see Figure S4a). As expected, the five $\alpha$-subpopulations displayed distinct green fluorescence signal distributions and different fractions of positive droplets (Fig. 3c). However, the quantitative influence of the mixing fraction $\alpha$ on the observed fraction of positive droplets was surprising. In our experiment, we intentionally used a high bacterial density, ensuring that the average number of bacteria in the droplets for all tested mixing ratios satisfied $\lambda_b > 15$. This leads to a situation where virtually every droplet contains at least one bacterium for all

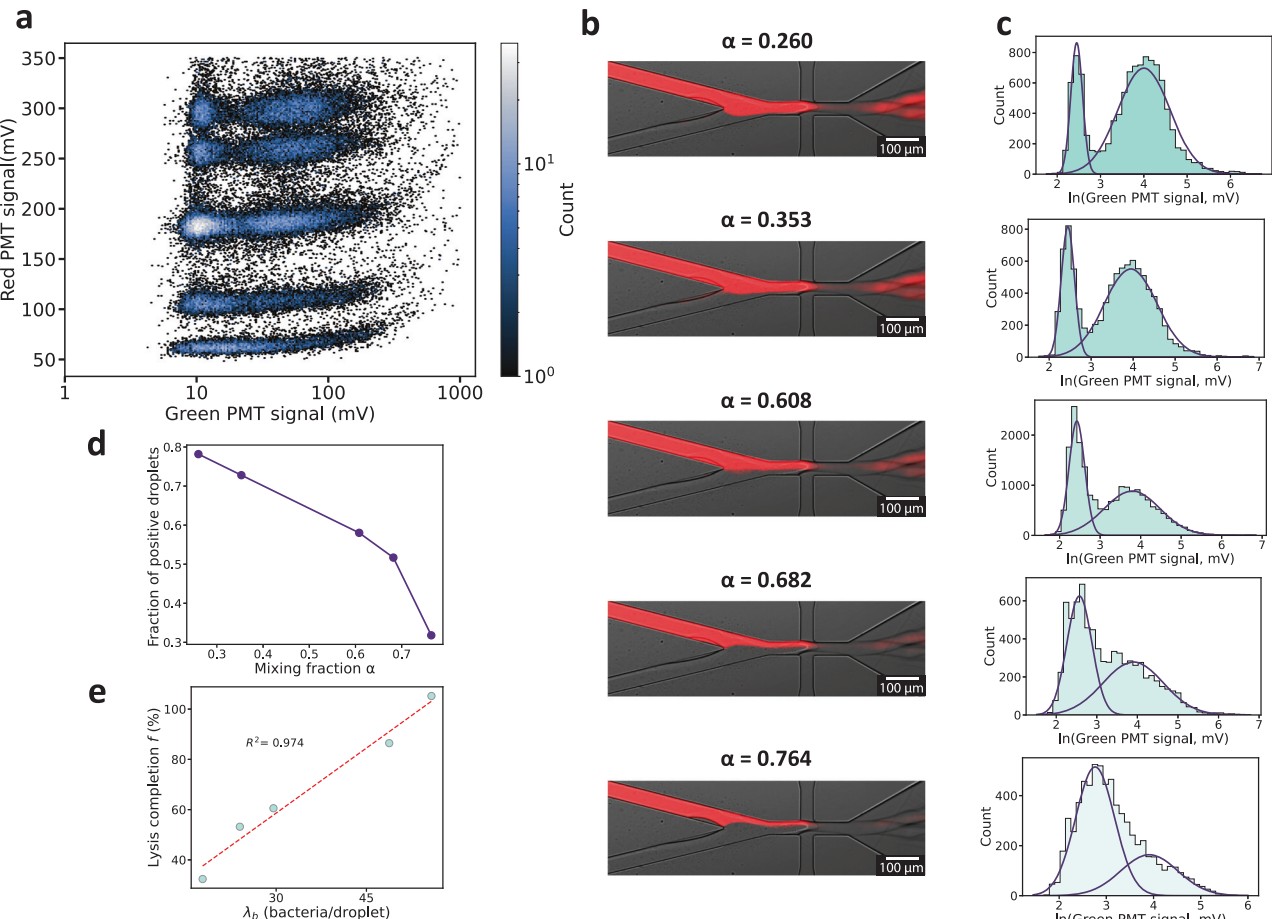

**Fig. 3 | Screening of discrete mixing fractions. a** 2D histogram plot of red and green fluorescence signals, $n$ = 6.40 × 10⁴ droplets. **b** Fluorescence and bright-field microscopy overlay image of the flow focusing junction during droplet generation. The determined average value of the mixing fraction $\alpha$ is indicated for each subpopulation. This protocol of droplet production was repeated in 4 independent experiments. **c** Logarithm-transformed green fluorescence signal distribution for each subpopulation (light blue bars). Purple curves represent the positive and negative droplet modes predicted by GMM with silhouette scores of 0.616, 0.609, 0.612, 0.605 and 0.609 and $n$ = 1.09 × 10⁴, 9.30 × 10³, 2.01 × 10⁴, 7.59 × 10³ and 6.86 × 10³ droplets for $\alpha$ = 0.260, 0.353, 0.608, 0.682, 0,764, respectively. **d** Fraction of positive droplets $P_L$ as a function of $\alpha$. **e** Lysis completion in function of the expected bacteria droplet occupancy $\lambda_b$. The 100% lysis completion fraction of positive droplets is computed from DLA titer and $\lambda_b$ from bacterial cell density in solution. dotted lines: linear fit.

mixing ratios $\alpha$. Consequently, the fraction of positive droplets $P_L$ is expected to depend solely on the number of phages encapsulated. As the expected number of phages per droplet, $\lambda_p$, is directly proportional to $\alpha$, droplets with a larger $\alpha$ were assumed to exhibit a higher fraction of positive droplets compared to those with a smaller $\alpha$. Contrary to this expectation, the fraction of positive droplets was observed to decrease with increasing $\alpha$, as shown in Fig. 3d.

To account for this result, we hypothesize that the number of encapsulated bacteria was lower than anticipated from the OD measurement of the initial bacterial suspension. This indicates that $\lambda_b$ should be interpreted as the effective number of bacteria capable of participating in a lysis reaction, excluding those that are dead or in a dormant metabolic state[44,45]. Furthermore, we suspect that the observed fraction of positive droplets also depends on the kinetics of the lysis reaction. Phage adsorption, and thereby lysis, occurs more slowly at lower bacterial densities[46], corresponding to higher mixing fractions $\alpha$. Thus, at high $\alpha$ fractions, the lysis reaction may not yet have been completed.

We quantified the degree to which the observed fraction of positive droplets, $P_L^{exp}$, deviated from the theoretically expected fraction, $P_L^{th}$, at the time of observation by defining the fraction of lysis completion $f = \frac{P_L^{exp}}{P_L^{th}}$. We computed $P_L^{th}$ using equation (1), where we used a phage titer $c_p$ measured via DLA, a cell density $c_b$ derived from

the optical density (OD) of the encapsulated bacteria solution, and the given values of the mixing fraction $\alpha$ and the droplet volume $V$. Notably, the parameter $f$ (Fig. 3e) appears to be linearly correlated with the expected number of bacteria per droplet. This hints at either an influence of the bacterial count on the speed at which lysis occurs, or an overall lower effective cell density.

**Screening with continuous mixing fractions**

We next sought to explore whether we could determine the effective bacterial density within the droplet experiment itself and thereby achieve a more accurate assessment of the digital titer. In particular, by continuously varying $\alpha$ rather than in discrete steps, we can obtain a large dataset of different $P_L$ values for various mixing fractions $\alpha$, which allows for the confident determination of both bacterial density $c_b$ and phage titer $c_p$ without reference to the bulk measurements. To this end, the mixing fraction was constantly changed during the emulsification process, resulting in monodisperse droplets with a large range of $\alpha$ values (see Figure S5). To avoid the potential influence of lysis kinetics on the results, bacterial density was substantially increased compared to the previous experiments.

We binned droplets according to their red fluorescence value, and linearly mapped these bins to specific $\alpha$ values, using the minimum and maximum $\alpha$ measured through microscopy as reference points

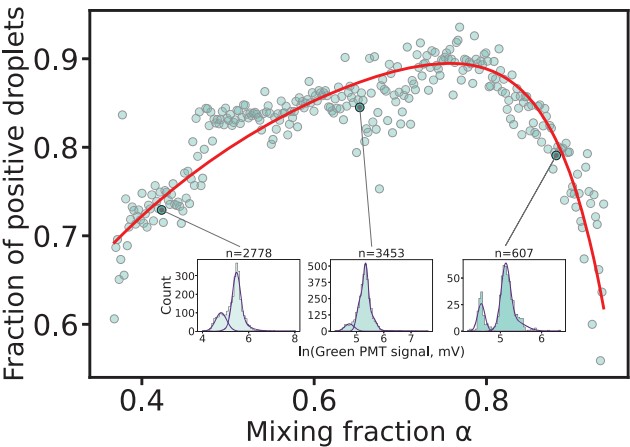

**Fig. 4 | Continuous screening of the mixing fraction.** Blue dots represent red fluorescence signal bins, containing each ≈ 2400 droplets on average. The total analyzed droplet population is $n = 7.20 \times 10^5$ droplets. Each bin is assigned to a $\alpha$ value according to its mean red fluorescence signal and its fraction of positive droplets $P_L$ is determined by GMM. Insets used here as examples, represent the logarithm-transformed green fluorescence signal values distribution (blue bars) for three bins across the $\alpha$ values range as well as the modes determined by GMM (purple curves), n is the number of droplets in the bin. The red curve represents a fit function of the equation (1) ($R^2 = 0.784$) leading to a phage titer of $1.205 \times 10^8$ PFU/mL and on OD of 0.773.

(see SI). For each bin, our Gaussian mixture model was used to determine $P_L$ which allowed us to determine its dependence on $\alpha$ (Fig. 4). Fitting our statistical model (Eq. (1)) for the fraction of positive droplets $P_L(\alpha)$ to the experimental data yielded a digital titer (see Fig. 4) of $1.205 \times 10^8$ PFU/mL, which is very close to the titer ($1.180 \times 10^8$ PFU/mL) determined by DLA, thus validating our approach as a viable alternative to the traditional method. Notably, the bacterial density obtained from the fit was only about 28.6% of that estimated from the bulk OD measurement. As discussed above, this effective concentration likely represents only those cells that are susceptible to infection, excluding dormant or metabolically inactive bacteria[47–52], as well as resistant cells and potentially dead bacteria or cellular debris that contribute to the optical density signal[44,45].

### Lysis kinetics in emulsions with a bimodal size distribution

An important advantage of our system is that it enables direct monitoring of lysis kinetics. We employed this capability to study the dynamics of phage infection by following co-encapsulated phages and bacteria over time. To assess the influence of droplet size on lysis kinetics, we determined the fraction of positive droplets for two distinct droplet volumes. By varying the pressure applied to the dispersed phase, we generated two distinct droplet populations within the same emulsion (Fig. 5a). The resulting populations substantially differed in mean diameter, but each in itself was monodisperse (Fig. 5b). The two populations and larger droplets resulting from droplet coalescence could be clearly separated from each other via the red reference fluorescence signal (Fig. 5c). Similar to emulsions containing subpopulations with different mixing fractions $\alpha$, the combined analysis of different droplet sizes enables an extension of the titer dynamic range (Figure S6).

The bimodal emulsions were analyzed after different incubation times, and for each subpopulation, the temporal evolution of the fraction of positive droplets was obtained (Fig. 5d). Both droplet sizes exhibited similar lysis kinetics, asymptotically approaching a plateau expected to correspond to the completion of lysis in all droplets containing both phages and bacteria. Consistent with this interpretation, the fraction of positive droplets derived from the DLA titer after overnight incubation closely matches this value.

To gain further insight, we developed a baseline model for phage infection kinetics in droplets (cf. Supplementary Information for details). In this model, phage adsorption follows second-order mass-action kinetics, with rate constant $k$, consistent with the commonly used bulk model[46,53–59]. A droplet is classified as positive in our model if at least one phage adsorbs to a bacterium. Furthermore, we incorporate a time delay $\tau$ between encapsulation and the detection of the first positive droplets, which can be interpreted as a maturation time[59,60]. Assuming a sufficiently large number of encapsulated bacteria, consistent with the experimental conditions, the fraction of positive droplets $P_L(t)$ at time $t$ is given by

$$P_L(t) = 1 - e^{-\alpha c_p V A(t)}, \qquad (2)$$

where

$$A(t) = 1 - e^{-(1-\alpha)kc_b(t-\tau)}. \qquad (3)$$

Here, the product $kc_b$ can be interpreted as an effective first-order rate constant[46]. Fitting the equation to the data allowed us to determine the phage titer, resulting in $8.30 \times 10^7$ PFU/mL and $8.46 \times 10^7$ PFU/mL for the smaller and larger droplets, respectively. The lag times were $\tau = 24.6$ min and $\tau = 34.5$ min, and the effective first-order rate constants were $kc_b = 0.0061$ min$^{-1}$ and $kc_b = 0.0050$ min$^{-1}$.

As before, the digital titers closely matched the DLA titer, which was determined to be $9.55 \times 10^7$ PFU/mL. The determined lag times $\tau$ are of the same order of magnitude as previously reported values for T7 phage maturation[41]. However, $\tau$ is also influenced by the emulsification process, which can take up to 10 min. Reported adsorption rate constants $k$ for T7 phage binding to *E.coli* range from $1 \times 10^{-10}$ mL min$^{-1}$ to $1 \times 10^{-9}$ mL min$^{-1}$ for different media[56]. With a measured bacterial density of $c_b = 1.83 \times 10^9$ cells/mL, and taking into account the effective bacterial fraction of 28.6% determined earlier, we computed adsorption rate constants of $k = 1.16 \times 10^{-11}$ mL min$^{-1}$ for the smaller droplets and $k = 9.50 \times 10^{-12}$ mL min$^{-1}$ for the larger ones. We attribute the significant deviation of the adsorption rate to the fact that we used PBS and not cell culture media in our experiment[56], and also to the imprecise determination of $c_b$ from optical density measurements. The time courses depicted in Fig. 5d indicate that droplet volume does not significantly affect the time scale of lysis. Notably, our model captures the effect of varying droplet sizes well, as reflected by the similar computed fit parameters obtained for the data from the two volumes.

Remarkably, the time required to lyse all lysis-competent bacteria differs considerably between the digital assay and DLA (Fig. 5e), with lysis completing much faster for the latter. In a double-layer agar plate assay, complete lysis is defined as the point at which every phage deposited in the agar produces a visible plaque. The more rapid completion of lysis in DLA likely reflects the favorable growth conditions within the nutrient-rich agar, where each phage is surrounded by metabolically active bacteria that strongly support phage infection dynamics[20,61].

## Discussion

In this work, we introduced a novel high-throughput technique based on droplet microfluidics that enables the study of phage infections in a digitized manner. Unlike existing methods, this technique can, in principle, be applied to any lytic phage-host pair. The droplet composition can be freely tailored, allowing systematic variation of parameters such as medium composition, microbial density, temperature, and exposure time. By encapsulating host-phage pairs at defined mixing ratios into hundreds of thousands of droplets within minutes, the method enables high-throughput, statistically robust analysis of individual phage infection events.

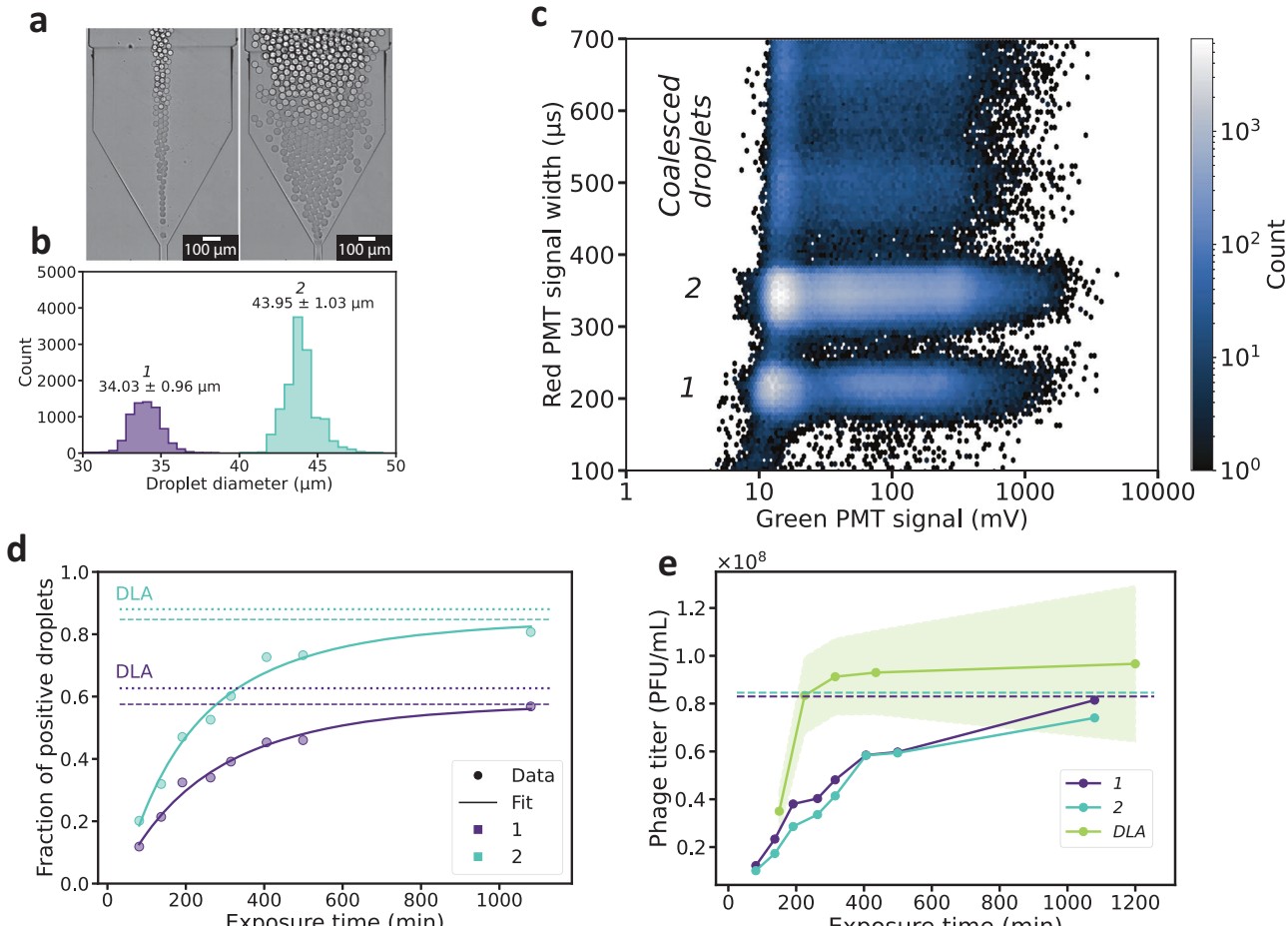

**Fig. 5 | Analysis of emulsions with a bimodal size distribution. a** Monitoring droplet generation by bright field microscopy. This protocol of emulsification with two size modes was repeated in 4 independent experiments. **b** Droplet size measurements. $n= 6.70 \times 10^3$ droplets and $n= 1.28 \times 10^4$ droplets for size mode 1 and 2, respectively. **c** 2D histogram plot of green fluorescence signals and red fluorescence signal width after 191 min of incubation, $n= 6.85 \times 10^5$ droplets. **d** Fraction of positive droplets from different exposure time of bacteria to phages for each size mode. Dots: data from droplet digitization. n= $1.32 \times 10^5$, $3.51 \times 10^4$, $1.36 \times 10^5$, $4.79 \times 10^4$, $2.73 \times 10^4$, $2.41 \times 10^4$, $6.10 \times 10^3$, $1.91 \times 10^5$ and $3.16 \times 10^5$, $1.22 \times 10^5$,

$4.90 \times 10^5$, $4.45 \times 10^4$, $6.54 \times 10^4$, $8.10 \times 10^4$, $7.10 \times 10^4$, $4.35 \times 10^5$ droplets for size mode 1 and 2, respectively. Curves: fit function of Equation (2). Dashed line: limit value from fit. Dotted line: computed fraction of positive droplets based on titer obtained from DLA. **e** Digital phage titer computed from different exposure times for each size mode and phage titer from DLA. Blue and purple curves show the digital titers for the individual size modes. The green curve represents the mean DLA titer (*n*=3, technical replicates), with the shaded green area indicating ± standard deviation across replicates. Dashed line: phage titer from fit for 2 size modes.

We demonstrated several variations of the method that provide comprehensive insights into phage-host interactions by digitizing individual infection events. The technique enables the accurate determination of phage titers. Modulating the mixing ratio of the phage and bacterial suspensions allows to adjust and expand the dynamic range of lysis experiments, covering a broad range of PFU values. Moreover, this approach provides more information than conventional emulsions with a uniform composition, which can be exploited to determine the effective titers of both participants of the lysis reaction - the host and the phage.

Notably, our system can be utilized to study phage infection kinetics at the single-event level. In conventional bulk assays, the study of infection kinetics is complicated by the fact that progeny phages from each burst are released to the environment and thus contribute to the infection of the remaining bacteria, masking the effect of the initial phage population. By contrast, encapsulation of phages and their hosts into small compartments allows counting discrete infection events, which can be attributed to the initially present phages. In other terms, infection kinetics in a bulk assay is dominated by the most infectious phages in the suspension and their progeny, while our digital assay gives insight into the diversity of phage infectivity within the population.

In the present study, we deliberately encapsulated bacteria in PBS, a non-nutrient buffer that does not support growth. This choice simplifies analysis, as the bacterial count within each droplet remains constant throughout the experiment. However, our system could also be operated with culture media in which the bacteria grow to their maximum density, which would reduce the time required for the completion of lysis across all droplets[20,61]. Moreover, our approach can, in principle, be scaled to droplet numbers of up to $10^7$, enabling the investigation of systems with highly unbalanced ratios of positive to negative droplets and thus allowing the detection of subtle differences in infectivity between phage-host pairs.

Finally, our droplet microfluidic setup could be integrated with a droplet sorting unit, enabling high-throughput screening and selection of phage variants of interest. Such capability would be particularly beneficial for applications in phage engineering and directed evolution, providing an approach to optimize therapeutic phages for enhanced infectivity and an extended host range. In conclusion, the digital phage experiments presented in this study allow the determination of multiple key infectivity parameters within a single experiment. This technique thus holds potential for diverse applications, ranging from fundamental

studies of phage biology to the development of next-generation therapeutic phages.

## Methods

### Preparation of microfluidic devices

Microfluidic chip designs were developed using AutoCAD software. Silicon master molds were created from 2-inch silicon wafers (Siegert Wafer, Germany), employing photolithography with SU8-3050 photoresist (Micro resist Technology) and a $\mu$MLA100 tabletop maskless aligner (Heidelberg Instruments). Subsequently, PDMS microfluidic devices were crafted via soft lithography. Specifically, 15 g of PDMS (Sylgard 184, Dow Corning) was thoroughly mixed with 1.2 g of curing agent, poured over a silicon master enveloped in aluminum foil, degassed for 30 min, and then cured in an oven at 80 °C for one hour. After curing, the PDMS was removed from the master, trimmed to size, and inlet and outlet holes were introduced using a biopsy punch. Glass slides were cleaned with 2% Hellmanex III (Hellma) and distilled water, followed by drying at 80 °C for 30 min. Finally, the PDMS devices and glass slides were treated with $O_2$ plasma (1 min, 20 sccm, 100W) for bonding, then further cured together at 80 °C for an hour.

### High-throughput production of droplets

Bacterial cell cultures were prepared via dilution (1/500) of a saturated pre-culture, incubated at 37 °C, 250 rpm in NZYCM medium for 2 hours, then centrifuged (3000 G, 5 min) and re-suspended at an adequate bacterial density in PBS buffer (Fisher Scientific), typically $OD_{600} > 2.5$. One solution containing either phages or bacteria was supplemented with 2 $\mu$M of a reference dye (Atto 565, Sigma-Aldrich). Both solutions were supplemented with 2 $\mu$M of YOYO-1 (Invitrogen). For continuous mixing fraction $\alpha$ screening, the concentrations YOYO-1 and Atto 565 were 1 and 4 $\mu$M, respectively. Microfluidic channel walls were treated with 1% trichloro(1H,1H,2H,2H-perfluorooctyl)silane (Merck) dissolved in FC-40 oil (Merck) for 1 min immediately prior to use. Microfluidic devices were operated using a 4-channel Elveflow OB1 controller for the aqueous phase. The continuous phase, consisting of 2% fluorinated surfactant dissolved in FC-40 oil (FluoSurf™, Emulseo) was injected with a syringe pump (TSE Systems) and a 1 mL glass syringe (BGB Analytik). Dispersed and continuous phases were connected to the chip with Tygon ND 100-80 tubing (Saint-Gobain, ID 0.5 mm). Aqueous solutions were co-flowed into a flow focusing junction along side with the dispersed phase. Droplet production was monitored using a 10X P-Apo air objective (NA 0.45) on a Nikon Ti-2E equipped with a SOLA SM II LED light source, a motorized stage and an Andor NEO 5.5 camera. Emulsions were recovered using a plastic syringe and incubated in the dark at 37 °C, 250 rpm. The details of emulsions, their production and their analysis can be found in the supplementary information.

### High-throughput droplet scanning

For characterization, droplets were re-injected with a syringe pump (TSE Systems) into a dedicated chip treated similarly as the production chip. Spacer oil was injected with a 4-channel Elveflow OB1 controller. A white light laser source (SuperK Extreme, NKT Photonics) was used to excite fluorescence in the droplets at 491 and 565 nm through a Zeiss LD Achroplan 40x/0.60 Corr. Ph 2 microscopy objective at various rates (0.1-2kHz). Spacing oil flow through a flow-focusing junction allows droplets to be excited individually. The fluorescence emission signal was measured by PMTs (H10722-20, Hamamatsu) and recorded with a FPGA DAQ card (PCIE7841R, National Instruments) configured with LabVIEW. Laser beam positioning on the droplets was achieved using a high-speed camera (Mikrotron, Germany), an LED light source (M700L4, Thorlabs) and a custom-made microscope stage.

### Preparation of phage stock solution

Bacteriophage stocks of T7 phage were obtained by phage propagation on the host *E.coli* DSM 613 in liquid medium. Briefly, a overnight culture of *E.coli* was diluted 1:100 into 50 mL of NZCYM Medium (Carl Roth) and incubated at 37 °C, 250 rpm in a shaking incubator to OD 0.6-0.8. The culture was then infected with $10^8$ PFU of T7 phage and incubated for 3-4 h until full lysis. The solution was centrifuged 10 min, 7660 g to pellet bacteria debris and the supernatant was sterile-filtered using a 0.45 $\mu$m CFME syringe filter. Phage solution was supplemented with 0.1% BSA, aliquoted and stored at 4 °C. The titer of the phage solution was determined regularly by DLA to account for degradation. For each droplet experiment, the most current titer was used as reference.

### DLA

A double-layer agar plaque assay was used for bulk determination of the phage titer. Liquefied top-agar (NCZYM, 0.75 % Agar) was mixed with 100 $\mu$l of bacterial culture in late-log or stationary phase and 100 $\mu$l of phages in varying dilutions, and the mixture was used to overlay bottom-agar (NZCYM, 1.5 % Agar). Plates were incubated at 37 °C for at least 4 h before evaluation. All DLA experiments were performed using at least 2 technical replicates. The titer (PFU/mL) was calculated as follows: $n \cdot d/V$, with $n$: number of plaques counted (mean of technical replicates), $d$: dilution factor of phage stock, $V$: volume of phage solution plated (100 $\mu$L). Pictures of petri dishes used for DLA with this protocol are provided in Figure S8. To determine titer development over time, DLA was performed with 2 different phage solution dilutions and plaques were counted at different timepoints. The titer for each timepoint was determined as mean of all replicates where plaques could be discerned. The absolute titer was determined as mean of all replicates from the latest timepoint where plaques could be distinguished for each of the dilutions.

### Bulk assay

Fluorescence emission intensity was measured in 384-well plates with a BMG CLARIOstar plate reader at 37 °C and shaken at 700 rpm. Cell and phage solutions were diluted in PBS. All samples contained 1 $\mu$M YOYO-1 dye and T7 phages were added at a nominal $1.35 \times 10^8$ PFU/mL.

### Ethics

This study does not involve experiment involving animals, human participants, or clinical samples.

### Reporting summary

Further information on research design is available in the Nature Portfolio Reporting Summary linked to this article.

## Data availability

All data supporting the findings of this study are available within the article and its supplementary files. Any additional requests for information can be directed to, and will be fulfilled by, the corresponding authors. Source data are provided with this paper. The raw data generated in this study have been deposited in the Zenodo repository and is accessible under (https://doi.org/10.5281/zenodo.11074304). Source data are provided with this paper.

## Code availability

Data analysis scripts are available in Zenodo at (https://doi.org/10.5281/zenodo.11074304).

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

## Acknowledgements

This project has received funding by the European Commission through its Horizon 2020 research and innovation program under the Marie Skłodowska-Curie grant agreement No. 813786 EVOdrops (F.C.S.). We gratefully acknowledge funding by the Bavarian Ministry of Economic Affairs, Regional Development and Energy through an m4 award (project M4-2110-0004 "In vitro Synthese multivalenter Bakteriophagen zur Therapie von antibiotika-resistenten Infektionen" (S.v.S.). This project was funded by the European Innovation Council through the project MI-DNA DISC - grant agreement no. 101115215 (F.C.S.). We are thankful to Prof. Baret and David Van Assche for their assistance in the conception of our high-throughput droplet scanning setup.

## Author contributions

L.G., S.v.S., F.K. conceptualised the project. L.G. designed and performed experiments, wrote analysis scripts and wrote the original draft. S.v.S. performed DLA and phage stock preparation. F.K. wrote phage kinetics analysis scripts. L.G., S.v.S., F.K., and F.C.S co-wrote the manuscript. F.C.S supervised the work.

## Funding

## Competing interests

The authors declare no competing interests.
