## [Transparent Peer Review file · Nature Communications]

Quantifying Phage-Host Dynamics using Droplet Microfluidics

Corresponding Author: Professor Friedrich Simmel

Version 0:

Reviewer comments:

Reviewer #1

(Remarks to the Author)

The authors reported a droplet-based microfluidics method to encapsulate phages and bacteria, along with a statistics-guided approach to analyze the lysis kinetics in the droplets. However, there have already been many publications on droplet-based microfluidics and other microfluidic techniques for phage quantification and lysis.

1. The authors need further demonstration of novelty, since the droplet formation method is well studied, and bacteriophage-bacteria interaction in droplets has also been investigated by other groups.

2. The title is too broad, it needs to be more specific and focused.

3. The authors utilize droplet microfluidics for bacteriophage studies, but their device does not demonstrate significant enhancements over existing methods for bacterial phage encapsulation. While the manuscript frequently highlights the advantages of high throughput, there is a lack of further elaboration and application regarding this aspect.

4. The author should consider real-world scenarios instead of merely calculating expected conditions. Some of the calculated conditions may not be suitable for actual phage experiments.

5. In the method part of DLA, "Plates were incubated at 37°C for at least 4 h until plaque formation was visible." Please provide the original phage plaque pictures from the double-layer plaque assay. Is a tenfold dilution used to calculate the phage number? Additionally, please indicate the number of replicates performed in this experiment for both droplets and DLA in the manuscript.

6. In Figure 2, it can be helpful to provide both bright-field and fluorescent images of digitized droplets to distinguish the negative and positive infection of phage. Please explain the threshold of the green PMT signal used to discriminate between positive and negative droplets, as there appears to be overlap between the signals of positive and negative droplets on the fitting curve. Additionally, why is there a 90-minute incubation period in Figure 2 and a 240-minute incubation period in Figure 3?

7. In Figure 3, what is the real dynamic range of mixing fraction for $\alpha = 0.260, 0.353, \text{ and } 0.608$, assuming the number of droplets is 10^6 ? In scenarios with larger mixing fractions ($\alpha > 0.608$), is there reliable and significant difference between positive and negative signals in the real experiment (such as 240-minute)?

(Remarks on code availability)

DOI NOT FOUND

10.5281/zenodo.11074305

This DOI cannot be found in the DOI System. Possible reasons are:

The DOI is incorrect in your source. Search for the item by name, title, or other metadata using a search engine.

The DOI was copied incorrectly. Check to see that the string includes all the characters before and after the slash and no sentence punctuation marks.

The DOI has not been activated yet. Please try again later, and report the problem if the error continues.

Reviewer #2

(Remarks to the Author)

This paper presents a method to quantify a phage using droplet microfluidics. A certain type of phage and host bacteria are encapsulated in single droplets with varied combinations containing various number of phages and bacteria. A red reference dye is added to indicate the volume fraction of the droplet from the phage sample. A green DNA staining dye is added to

count the droplets in which the bacteria is lysed which also means at least one phage is presented. The fraction of the droplets containing phages is obtained by an equation that can predict the probability that a droplet contains at least one phage and one bacteria, when bacteria is in excess. The authors also investigate the infection kinetics by monitoring the fraction of positive droplets with phage exposure time.

The microfluidic setup using flow focusing design to generate droplets is quite matured with over 20 years' development and applications, no much to comment. The application of using populated droplets with bacteria and phages to quantify the phages with statistical analysis is interesting. However, there seems quite a few logical loopholes in the analysis.

1. The whole analysis is based on an assumption as mentioned in the section "Statistics of bacterial lysis in droplets", (first paragraph, line 3-5), that "... every encapsulated phage will lead to the lysis of a bacterium when on is available". That means a single copy of phage is considered to lead to a positive droplet. However, is that the real case? Is it possible that only a percentile of phage is active to infect bacteria, just like only a percentile of bacteria can be infected by a phage as demonstrated in this paper? Without a strong theoretical or empirical evidence to solidify this assumption, the whole analysis in this paper would collapse.

2. In Fig. 2, the peak position corresponds to the green fluorescence intensity, while the peak height corresponds to the number of droplets of a certain fluorescence intensity. The green intensity is from the YOYO-1 dye to stain DNA from bacteria. However, the peak position of sample with neither bacteria nor phages in Fig. 2f is about 5, while the peak of bacteria alone in Fig. 2e is about 2. Why would a sample without any bacteria or phages have such a high fluorescence intensity? Considering the fluorescence intensity in the figures is already the $\ln(\text{intensity})$, the raw intensity would be around 20 times higher without any bacteria or phages than with bacteria. Why would this happen?

3. Fig. 2c, the negative peak is at around 3.8, which is similar to the positive peak in Fig. 2a. Here the authors restrict the fitting of data into two peaks. However, as can be seen in Fig. 2a, a shoulder with small bumps at higher intensity is seen. If we assume some droplets contained a lot of bacteria, more peaks can be fitted. Then back to Fig. 2c, is it possible the negative peak would actually be a positive peak, and there is few droplet with 0 phages in? For an unknown sample without knowing the phage concentration, how would you exclude this possibility?

4. Still in Fig. 2, Fig. 2b and Fig. 2d were to obtain the fraction of positive droplets in relation with phage titer based on Eq. 1, which depends on the of bacteria concentration (λ_b) and phage concentration (λ_p). However, based on the later sections, the authors found that not all bacteria is lysed by phages ("Screening with discrete mixing fractions" and "screening with continuous mixing fractions"). That only a percentile of bacteria are infected with phages, and the percentile depends on the bacteria concentration, as shown in Fig. 3e. If so, what λ_b was applied in the analysis of Fig. 2b and Fig. 2d? How to rationale what number to apply when another bacteria is used?

5. On page 8, first paragraph, the authors attribute the discrepancy of obtained alpha to the designed alpha to that "phage adsorption occurs more slowly at lower bacteria densities". If it was because of the slow adsorption rate, would this issue to be solved by longer culture time? As shown in Fig. 5, the fraction plateau after 1000 minutes, which is around 16 hours. Would a 16-hour culture solve the problem?

6. On page 9, paragraph 2, line 7, "the effective bacterial density determined via the fit was found to correspond to only 16.6% of that calculated from an OD measurement in bulk." Only 17% of bacteria is in active state seems quite low. Is there any literature or other experiment to support this low number?

(Remarks on code availability)

Reviewer #3

(Remarks to the Author)

The manuscript of Givélet et al describes the application of droplet microfluidics to monitor phage-host bacteria interactions. The approach allows the quantitation of individual phage infection events. Overall the manuscript is well written and opens new possibilities for the phage host interaction studies.

I don't have any critical comments and noticed just a couple of typos in the text or references.

Page 13, line 4 from bottom. Hammatsu written with 3 m's

Ref 6, the last author should be Harald Brüssow

(Remarks on code availability)

I am not competent to evaluate the code

Version 1:

Reviewer comments:

Reviewer #1

(Remarks to the Author)

While the authors have clarified the number of replicates and the titer calculation formula, their response regarding the original DLA data remains unsatisfactory. The claim that the lack of an automated plaque counter prevented the capture of any original images is unconvincing, as standard laboratory practice involves photographing plates for documentation, regardless of the counting method. Providing representative images from different experiments is not an acceptable substitute for the primary data generated in this study. Furthermore, the reported 4-hour incubation period for visible plaque formation is highly atypical and raises concerns about the validity of the DLA results used for comparison. Therefore, we

must insist that the authors provide the original raw data for the DLA assays. Specifically, how were the results recorded on the day of the experiment? A laboratory notebook entry or data sheet detailing the dilutions made, replicates, and incubation start time. The original plate count records. This is typically a table with columns for: Dilution Factor, Plate Identifier (e.g., Replicate A, B), Plaque Count, and Notes. Without this primary data, the foundation for comparing and validating their new method is significantly weakened.

I appreciate the authors' clarification on the principle of extending the dynamic range by varying the mixing fraction (α). However, the response does not fully address the specific question raised. The question specifically asked for the real dynamic range for $\alpha = 0.260, 0.353, \text{ and } 0.608$, assuming 10^6 droplets. The reply provides a plausible overall range (1.2×10^6 to 2.0×10^8 PFU/mL) achieved by varying α , but it fails to specify the effective range for each of the three individual α values used in Figure 3. To properly evaluate the method's performance and the data presented in Figure 3, it is essential to know the accurate dynamic range for each specific condition tested. The 5% to 95% positive droplet threshold is a reasonable criterion. Please provide a calculation or estimation of the upper and lower detection limits for each specific α value (0.260, 0.353, 0.608) based on this criterion and the stated total droplet number. This quantitative information is crucial for assessing the practical utility and precision of the technique for a given setup.

(Remarks on code availability)

I am not capable to evaluate the code.

Reviewer #2

(Remarks to the Author)

The authors have addressed all my concerns with more references and discussions. The current version looks good to me. I have no other comments.

(Remarks on code availability)

Version 2:

Reviewer comments:

Reviewer #1

(Remarks to the Author)

The authors have properly addressed the reviewers' concerns, it can be considered for publication.

(Remarks on code availability)

REVIEWER COMMENTS

We would like to thank our three reviewers for their critical assessment of our work, which helped us to considerably improve the presentation of our results and emphasize the significance of our methodology. A detailed point-by-point response is given below.

Reviewer #1 (Remarks to the Author):

The authors reported a droplet-based microfluidics method to encapsulate phages and bacteria, along with a statistics-guided approach to analyze the lysis kinetics in the droplets. However, there have already been many publications on droplet-based microfluidics and other microfluidic techniques for phage quantification and lysis.

1. The authors need further demonstration of novelty, since the droplet formation method is well studied, and bacteriophage-bacteria interaction in droplets has also been investigated by other groups.

REPLY: We are aware that some aspects of our work were present in previous works, such as the use of DNA intercalation as a readout or the use of lysis event detection in droplets as a basis for enumeration (as opposed for instance, to DNA template detection with digital PCR). However, none of the previously reported studies combines all of these elements within a single workflow, and our work introduces innovative components, particularly in the mathematical analysis we present here, which has not been reported before. Below, we list the key differences between other reported methods and our work, highlighting the innovative aspects of our approach.

- First of all, to our knowledge many aspects of our work have not been reported by other groups before. For instance, the co-flowing of the phage-containing solution and the bacteria-containing solution to enable a phage exposure exactly at the time of encapsulation has not been done before. Already published methods are all based on the encapsulation of a single solution containing phages and host. This clearly has the disadvantage of phage-host interaction starting before encapsulation, therefore introducing errors in phage quantification and infection dynamics monitoring.
- The use and analysis of different droplet sizes or mixing fractions within the same emulsion has not been previously reported, despite offering clear advantages. These include the ability to expand the dynamic range and to reveal trends regarding the influence of the phage-to-bacteria ratio or droplet volume on infection dynamics. Varying the mixing fraction also provides a means to determine the phage titer with greater confidence. For these reasons, we believe that our work demonstrates a more advanced use of droplet microfluidics for studying phage infection dynamics.
- Another aspect unique to our work is the sheer number of droplets used for quantification. Our system routinely analyses 10^5 droplets per experiment and leverages this large number to build robust statistics. This is made possible by the use of relatively small droplet volumes (compared to most studies of a similar nature), which also results in significant savings on reagents. Such throughput is also greatly facilitated by our detection system, which is able to

acquire up to thousands of droplets per second, compared to most of the similar studies employing microscope image analysis.

- Methods based on DNA template detection, such as digital PCR, tend to overestimate PFUs by detecting free DNA copies, non-infectious particles and phages outside the host range. This can lead to discrepancies of several orders of magnitude between the DNA copy number and the actual PFU count. Our method directly relies on the detection of lysis events, involving only host cells and phages that participate in the lysis process, which is also the basis of DLA.
- The dynamics of phage infection were previously studied in a bulk environment, but kinetic studies and subsequent modelling of phage lysis in droplets was, to our knowledge, never reported. Studying phage infection dynamics in droplets rather than in bulk allows to simplify the assumptions behind the mathematical model: in droplets, the phage progeny does not influence the outcome of the lysis in the droplet population, enabling a more straightforward modelling approach, as our results demonstrate.
- Lastly, the majority of studies similar to our work assume non-limiting bacterial encapsulation by employing high cell densities, thereby imputing a lysis event solely to the encapsulation of one or more phage particles. Our theoretical framework encompasses the encapsulation statistics of both host and phage, allowing us to work with any bacterial density and, in some cases, to quantify the number of bacterial cells involved in phage interactions. A specific advantage of working with any cell density is that, in the event of access to only a low bacterial cell density (for instance, due to slow growth), our method can still operate reliably.

We therefore firmly believe that our work shows a more advanced use of droplet microfluidics to study and enumerate bacteriophages: we did not only evidence a phage enumeration through digital analysis well correlated with gold standard technic in the field (DLA), but our theoretical framework included phages and bacteria shared Poisson statistics and kinetic modelling. We also devised ways to expand the dynamic range of our method, set the bases for kinetic modelling in droplets.

Below, we outline the main differences and similarities between the most comparable published scientific articles, to the best of our knowledge, and our work.

Reference	Similitudes with our work	Differences with our work
Hoshino, Miu, Yuri Ota, Tetsushi Suyama, Yuji Morishita, Satoshi Tsuneda, and Naohiro Noda. 2023. "Water-in-Oil Droplet-Mediated Method for Detecting and Isolating Infectious Bacteriophage Particles	 • Readout based on YOYO-1 intercalation • Similar throughput • Droplet sizes are similar to our work. 	 • Studying T2 not T7 phages. • Mainly aiming at detecting and isolating phages but proportion of fluorescent droplets is never used for phage enumeration. • Fixed droplet volumes

via Fluorescent Staining.” Frontiers in Microbiology 14 (December): 1–10. https://doi.org/10.3389/fmicb.2023.1282372.		 • Fixed mixed fraction (single solution encapsulation) • Phage exposure starts before encapsulation (single solution encapsulation) • Fluorescent droplet fraction is determined with an arbitrary threshold while ours relies on a GMM modelling, which is more reliable, less user dependent and automated.
Wang, Min S., and Nitin Nitin. 2014. “Rapid Detection of Bacteriophages in Starter Culture Using Water-in-Oil-in-Water Emulsion Microdroplets.” Applied Microbiology and Biotechnology 98 (19): 8347–55. https://doi.org/10.1007/s00253-014-6018-7.	 • Phage infection detection is based on fluorescence signal • Signal based on DNA labelling • T7 and E. coli is employed as a phage-host pair. 	 • This study describes a method to merely detect phages qualitatively but not to enumerate them. • Phage exposure starts before encapsulation (single solution encapsulation) • Using Propidium Iodide (PI) and not YOYO-1. YOYO-1, unlike PI exhibit a major fluorescence emission increase upon binding to DNA, thus yielding a much better signal to noise ratio. • Double emulsions (w/o/w) are used here whereas we employed single emulsion (w/o). • Double emulsions (w/o/w) are produced from single emulsions (w/o) by shaking . This method does not allow to control the droplet sizes and suffers from a lack of monodispersity. This lack of monodispersity then also affect the

		accuracy of computed titers.  • Droplet volumes sizes are much larger than in our work (diameter: 152 μm). • Droplet signal acquisition done through microscopy image analysis. • Droplet signal acquisition done through microscopy image analysis. • Fixed droplet volumes • Fixed mixed fraction (single solution encapsulation) • The achievable throughput in this study is much lower than in our work: the authors mention around 1060 droplets analysed. • Droplets compositions are much more complex than in our work. They involve polyglycerol polyricinoleate, whey protein isolate, bile salts... Our setup requires only a commercially available oil supplemented with surfactant.
Tjhung, Katrina F., Sean Burnham, Hany Anany, Mansel W. Griffiths, and Ratmir Derda. 2014. "Rapid Enumeration of Phage in Monodisperse Emulsions." Analytical Chemistry 86 (12): 5642–48.	 • Different droplet sizes employed • Phage infection detection is based on fluorescence signal 	 • Detection method based on recombinant phages, therefore strongly limited on applications. • Phage exposure starts before encapsulation (single solution encapsulation)

https://doi.org/10.1021/ac500244g.		 • Droplet sizes are much larger than in our work (diameter: 165 μm). • Fixed mixed fraction (single solution encapsulation) • Droplet signal acquisition done through microscopy image analysis. • Droplet acquisition throughput much lower than in our work (maximum 400 droplets per frame) • Fixed droplet volumes • Fixed mixed fraction (single solution encapsulation)
Li, Xiang, Qixin Hu, Xu Liu, Jianfang Liu, Nannan Wu, and Feng Shen. 2023. "Rapid Bacteriophage Quantification by Digital Biosensing on a SlipChip Microfluidic Device." Analytical Chemistry 95 (22): 8632–39. https://doi.org/10.1021/acs.analchem.3c01066.		 • Droplet signal acquisition done through microscopy image analysis. • Phage exposure starts before encapsulation (single solution encapsulation) • Droplet acquisition throughput much lower than in our work (2304 droplets per chip) • Droplet sizes are much larger than in our work (diameter $>100 \mu\text{m}$). • Phage infection detection is based on bacterial growth assessment.
Nikolic, Nela, Vasileios Anagnostidis, Anuj Tiwari, Remy Chait, and Fabrice Gielen. 2023. "Droplet-Based Methodology for Investigating Bacterial Population Dynamics in	 • Phage infection detection is based on fluorescence. • Droplet sizes are similar to our work. 	 • Detection method based on recombinant bacteria, therefore limited on applications. • Phage exposure starts before encapsulation

Response to Phage Exposure.” Frontiers in Microbiology 14 (November): 1–12. https://doi.org/10.3389/fmicb.2023.1260196.		(single solution encapsulation)  • Droplet signal acquisition done through microscopy image analysis. • Fixed droplet volumes • Fixed mixed fraction (single solution encapsulation)
Morella, Norma M., Shangyang Christopher Yang, Catherine A. Hernandez, and Britt Koskella. 2018. “Rapid Quantification of Bacteriophages and Their Bacterial Hosts in Vitro and in Vivo Using Droplet Digital PCR.” Journal of Virological Methods 259 (December 2017): 18–24. https://doi.org/10.1016/j.jviromet.2018.05.007.	 • High-throughput enumeration of phages. 	 • Detection method is based on DNA copy detection. This method, albeit straightforward is also detecting free DNA copies and non-infectious phage particles leading to several order of magnitude difference between PFUs and DNA copies. Authors mention that this difference is mainly due to the proportion of non-infectious phage particles, depending strongly on the phage strain.

These publications were already cited in the manuscript, but we now emphasize the novelty of our contribution more clearly.

2.The title is too broad, it needs to be more specific and focused.

REPLY: We changed the title to be more specific: “Quantifying Phage-Host Dynamics using Droplet Microfluidics”

3.The authors utilize droplet microfluidics for bacteriophage studies, but their device does not demonstrate significant enhancements over existing methods for bacterial phage encapsulation. While the manuscript frequently highlights the advantages of high throughput, there is a lack of further elaboration and application regarding this aspect.

REPLY: We emphasize that our system is the only reported method combining enumeration based on lysis event detection – which is more precise than digital PCR (dPCR) – and which has applicability to

any phage or host. In contrast, other methods employing lysis event detection rely on recombinant phages or hosts, which significantly limits their applicability.

Additionally, our system relies on a novel approach to bacteriophage encapsulation that has not been reported previously: bacteria and bacteriophages are flushed in separately and mixed only during the encapsulation process, thereby ensuring their co-encapsulation at a defined time point. This approach ensures that every enumerated phage was initially present in the solution and did not originate from a recent phage burst. It also enables precise timing of phage exposure, providing a convenient and robust method for studying infection dynamics.

In contrast, previously reported systems rely on encapsulating a pre-mixed solution of phages and bacteria. This distinction is critical, as such methods inevitably introduce errors and imprecision. Specifically, the uncontrolled exposure of bacteria to phages prior to encapsulation can lead to infection processes initiating prematurely. This obscures the exact timing of interactions, undermines the reliability of infection dynamics studies, and can result in highly inaccurate titer measurements if a burst event occurs before encapsulation. Operating microfluidic systems can be challenging and any mishap after the onset of exposure can have critical consequences for the outcome of the experiment in such systems.

The simultaneous encapsulation of two laminar streams introduces a unique parameter in this method, namely the mixing fraction, α . This parameter can be adjusted by experimenters to access a broader dynamic range for phage titer enumeration, providing an additional advantage to our approach. Furthermore, our theoretical framework for enumeration accounts for both the α parameter and the bacterial encapsulation probability. To our knowledge, this has not been reported before, as existing phage enumeration methods typically require a high host density to ensure that phage presence is the only limiting factor for lysis. With our framework, it becomes possible to work with low bacterial densities (e.g., when growth rates are slow) or even to measure the proportion of hosts participating in lysis.

The high-throughput droplet generation and data acquisition enable the collection of robust statistical data for enumeration. Most previously reported methods involve emulsions containing several orders of magnitude fewer droplets compared to our approach. This is primarily because most similar methods rely on microscopic image analysis, which requires substantial memory and computational resources, thereby limiting the throughput and the total number of droplets that can be acquired. In contrast, our method, which relies on PMT signal acquisition, is considerably more straightforward, more sensitive and inherently high-throughput. Because even highly unbalanced proportions can be confidently determined with very large numbers of droplets, this capability allows operation at the edge of the dynamic range, ensuring phage solution enumeration with high confidence.

In this proof of principle, we focused on acquiring large datasets for a given phage solution. However, an alternative strategy could involve generating fewer droplets for each solution to be titered. For example, a few thousand droplets (comparable to the number typically used in previously reported methods) could be generated and analyzed with our method in an extremely short time. The generation and acquisition of such small emulsions would theoretically take less than a minute (with droplet generation and analysis both operating in the kHz range), enabling the rapid sampling of multiple solutions.

Finally, a common limitation of phage titration is obtaining sufficient cell mass to either perform a DLA or, when enumeration is done with droplets, to reach a sufficiently high cell concentration to ensure

host cell encapsulation in all droplets. In our case, having a theoretical framework that accounts for host cell encapsulation allows us to identify droplets that are negative due to the absence of host cells, thereby enabling titration at very low cell concentrations. This feature is particularly relevant for very slow-growing pathogens.

We elaborate on these aspects more extensively in the revised version of the manuscript (see also reply to 1).

4. The author should consider real-world scenarios instead of merely calculating expected conditions.

REPLY: We are not sure whether the reviewer refers to the use of the term “expected number” in our manuscript. This refers to the statistical expectation value (mean) in the droplets derived from the bulk measurement. In our work, we systematically used the standard method for phage enumeration, the Double Layer Plaque Assay (DLA), to validate our approach and compare the digital titer with results obtained using the standard method. However, we would like to emphasize that our method is fully applicable without prior knowledge of the “expected” values, such as those provided by DLA. The Poisson parameters λ_p is obtained from a fit and then converted into the digital titer. In the case of varying α , we can additionally determine λ_b . This is now explicitly highlighted in the revised version of the text.

In practical terms, experimenters would not need to perform DLAs to enumerate phages in solutions and could instead rely solely on our method. Notably, our method is compatible with any phage-host pair and does not require the use of recombinant phages or bacteria, as is necessary in some previously reported methods. Since bacterial exposure to phages begins only after encapsulation, our protocol eliminates the urgency seen in other methods relying on lysis event detection. In those methods, encapsulation must be performed promptly after mixing hosts and phages, which introduces a time-sensitive step. This constraint is technically challenging, particularly given common microfluidic issues that can delay emulsification and lead to significant inaccuracies, as mentioned in response to comment #3.

Given the applicability of our system to potentially any phage-host pair, it is well-suited for clinical applications. Estimating titers with high statistical confidence is a valuable asset in clinical workflows. Additionally, our method offers mechanisms to expand its dynamic range, either through adjustments to the mixing fraction or variations in droplet volume. This flexibility is particularly useful in clinical scenarios where the titer rough estimate for a given sample is either unknown or imprecise. Applying our method systematically with dynamic range expansion could be especially effective for determining the titer of medical samples. Moreover, our method can estimate the proportion of hosts that are resistant or do not contribute to lysis (as shown in Figure 4) for a given phage, providing valuable insights for medical applications. This capability could assist in addressing phage resistance or optimizing phage-host pair matching for therapeutic use. As mentioned in response to comment #3, one potential application of our method is the generation of small emulsions (a few thousand droplets) for each solution to be titered, thereby maximizing the throughput of solutions processed. This approach is particularly well-suited for medical analysis platforms, where a large number of samples must be titered in the shortest possible time.

5. In the method part of DLA, “Plates were incubated at 37°C for at least 4 h until plaque formation was visible.” Please provide the original phage plaque pictures from the double-layer plaque assay. Is

a tenfold dilution used to calculate the phage number? Additionally, please indicate the number of replicates performed in this experiment for both droplets and DLA in the manuscript.

REPLY: We do not possess the original pictures of DLA since our laboratory is not equipped with an automatic plaque counter. However, the counting of plaques was un-ambiguous and could hardly lead to misinterpretation. To support this claim, we now included pictures of DLAs carried out in an identical fashion. All DLA experiments were performed using at least 2 technical replicates. The titer was calculated as follows:

$$Titer = n \times d \times 10$$

With:

Titer the phage titer in PFU/mL

n the number of plaques counted (mean of technical replicates),

d the dilution factor of phage stock

10 is the dilution factor arising from using 100 μ l = 1 mL/10 of diluted phage solution in the DLA.

6. In Figure 2, it can be helpful to provide both bright-field and fluorescent images of digitized droplets to distinguish the negative and positive infection of phage.

REPLY: In response to your comment, we have added bright-field images of emulsification and droplet generation in Figure 1 to help readers better visualize the experimental workflow. Our setup, however, does not record fluorescence images of droplets during analysis, as this approach would be too slow for high-throughput data acquisition. Instead, fluorescence signals are detected exclusively by a photomultiplier tube (PMT), which is inherently more suitable for rapid measurements.

To address your request, we explored obtaining time-lapse fluorescence images on an alternative optical setup. However, capturing such data at the relevant scale poses substantial technical challenges, particularly because it requires stable droplet immobilization. Since our workflow does not rely on fluorescence microscopy images for analysis, we ultimately decided against including such data, so as not to give readers a misleading impression of our methodology.

Please explain the threshold of the green PMT signal used to discriminate between positive and negative droplets, as there appears to be overlap between the signals of positive and negative droplets on the fitting curve.

REPLY: Our data analysis workflow relies on distinguishing positive and negative droplets to compute the digital titer based on their respective proportions. However, as pointed out, the fluorescence signal difference between positive and negative droplets is often insufficient to separate subpopulations using a simple threshold. In most cases, subpopulations can be identified by their local maxima but exhibit significant overlap, as their distributions are not fully resolved.

To address this, we employ a Gaussian Mixture Model (GMM) to distinguish subpopulations despite overlaps. This technique dynamically fits multiple distributions to the data, enabling effective, user-independent separation. While our initial approach relied on an arbitrary threshold, we found it to be overly user-dependent and therefore potentially biased, and hence unsuitable for cases with substantial subpopulation overlap. This is now explicitly stated in the revised text.

A technical detail of our analysis workflow is that droplet populations are better described by log-normal distributions rather than normal ones. Therefore, we apply the GMM to logarithmically

transformed data. Overall, this approach renders our digitization workflow nearly fully automated, user-independent, and capable of resolving highly overlapping subpopulations. Moreover, it enables digitization even when positive droplets consist of multiple subpopulations with highly variable DNA content, such as those arising from different host densities or infection cycles.

Additionally, why is there a 90-minute incubation period in Figure 2 and a 240-minute incubation period in Figure 3?

REPLY: The emulsions shown in Figure 2 represent a proof-of-concept and illustrate the most straightforward application of our method, without variation in the mixing fraction (α). In contrast, Figure 3 demonstrates a more advanced application of our method, where droplet subpopulations with different mixing fractions are generated within the same emulsion.

For droplets with very high α values (and consequently very low expected bacterial densities per droplet), we anticipated that a longer time might be required to complete lysis. To account for this, the overall incubation time was increased to ensure that lysis was fully completed even in droplets with high α values. This is now specifically mentioned in the SI caption for Table 2.

7. In Figure 3, what is the real dynamic range of mixing fraction for $\alpha = 0.260, 0.353,$ and $0.608,$ assuming the number of droplets is 10^6 ?

REPLY: In Figure 3, as in other figures, we did not calculate an exact dynamic range for specific parameters. Instead, we show in Figure S4a that higher mixing fractions push the dynamic range towards lower values, whereas lower mixing fractions shift it towards higher values. This occurs because changing the mixing fraction artificially dilutes or enriches the final droplet composition in phages.

To determine the precise boundaries of a dynamic range, one must identify the thresholds above or below which the fraction of positive droplets is not sufficiently well correlated with the phage titer to provide a reliable titer estimation. In order to estimate a plausible dynamic range, for a bimodal droplet signal distribution, we considered that discrimination between 5% and 95% of droplets, remains feasible (even more unbalanced proportions may be possible to discern though). As illustrated in Figure S4a, by varying the mixing fraction alone (in the same proportion that in Figure 3) the detection range stretches across 1.2×10^6 to 2.0×10^8 PFU/mL. This shows that changing the mixing fraction provides a convenient means to increase the dynamic range (in both directions) among other ways such as adapting the droplet diameter or diluting phages solutions. We replaced the Figure S4a in the Supporting Information by the figure shown below.

In scenarios with larger mixing fractions ($\alpha > 0.608$), is there reliable and significant difference between positive and negative signals in the real experiment (such as 240-minute)?

REPLY: Emulsions with mixing fractions of 0.682 and 0.764 yielded clearly separable droplet populations, with silhouette scores of 0.605 and 0.609, and markedly different positive fractions (0.517 vs. 0.318). The apparent balance of populations in Figure 3c is deceptive, but the Gaussian Mixture Model allowed us to extract their true proportions in a user-independent manner.

Reviewer #1 (Remarks on code availability):

DOI NOT FOUND

10.5281/zenodo.11074305

This DOI cannot be found in the DOI System. Possible reasons are:

The DOI is incorrect in your source. Search for the item by name, title, or other metadata using a search engine.

The DOI was copied incorrectly. Check to see that the string includes all the characters before and after the slash and no sentence punctuation marks.

REPLY: We apologize for the broken link. The zenodo repository is now correctly linked via the given DOI.

Reviewer #2 (Remarks to the Author):

This paper presents a method to quantify a phage using droplet microfluidics. A certain type of phage and host bacteria are encapsulated in single droplets with varied combinations containing various number of phages and bacteria. A red reference dye is added to indicate the volume fraction of the droplet from the phage sample. A green DNA staining dye is added to count the droplets in which the bacteria is lysed which also means at least one phage is presented. The fraction of the droplets containing phages is obtained by an equation that can predict the probability that a droplet contains at least one phage and one bacteria, when bacteria is in excess. The authors also investigate the infection kinetics by monitoring the fraction of positive droplets with phage exposure time.

The microfluidic setup using flow focusing design to generate droplets is quite matured with over 20 years' development and applications, no much to comment. The application of using populated droplets with bacteria and phages to quantify the phages with statistical analysis is interesting. However, there seems quite a few logical loopholes in the analysis.

1. The whole analysis is based on an assumption as mentioned in the section "Statistics of bacterial lysis in droplets", (first paragraph, line 3-5), that "... every encapsulated phage will lead to the lysis of a bacterium when one is available". That means a single copy of phage is considered to lead to a positive droplet. However, is that the real case? Is it possible that only a percentile of phage is active to infect bacteria, just like only a percentile of bacteria can be infected by a phage as demonstrated in this paper? Without a strong theoretical or empirical evidence to solidify this assumption, the whole analysis in this paper would collapse.

REPLY: The reviewer is correct in pointing out that not every encounter between a phage and a host cell may result in lysis. Our original phrasing — "... every encapsulated phage will lead to the lysis of a bacterium when one is available" — was indeed misleading. In reality, only a fraction of phages in contact with host cells successfully initiate infection and cause lysis. This proportion depends primarily on the phage's affinity for the host, but is also influenced by factors such as the multiplicity of infection (MOI), the metabolic state of the host, and the surrounding physico-chemical conditions. We now write "every infective phage" in the revised version of the paper and clarified the statement. Since our detection method — actually like DLA and most other enumeration techniques — is based on observing lysis events, it inherently measures only the *active* phage population, i.e., those phages capable of infecting and lysing host cells under the given experimental conditions. Similar to other methods, our approach is designed to quantify and analyze only the active fraction of phages within a specific infectious context (e.g., host specificity, physicochemical conditions), rather than the total phage population present in the sample. This active fraction represents the most relevant metric for characterizing phage activity. The validity of our theoretical framework is supported by the close agreement between the titers obtained with our method and those measured by DLA.

2. In Fig. 2, the peak position corresponds to the green fluorescence intensity, while the peak height corresponds to the number of droplets of a certain fluorescence intensity. The green intensity is from the YOYO-1 dye to stain DNA from bacteria. However, the peak position of sample with neither bacteria nor phages in Fig. 2f is about 5, while the peak of bacteria alone in Fig. 2e is about 2. Why would a sample without any bacteria or phages have such a high fluorescence intensity?

Considering the fluorescence intensity in the figures is already the $\ln(\text{intensity})$, the raw intensity would be around 20 times higher without any bacteria or phages than with bacteria. Why would this happen?

REPLY: Indeed, the signal in Figure 2f is much higher compared to the other panels, despite the fact that the droplets do not contain bacteria or phages. This results from variations in parameters that affect the PMT signal, primarily the PMT gain, the focus, and the laser source power. We emphasize that our method does not rely on absolute fluorescence levels to distinguish negative from positive droplets, but rather on the relative fluorescence levels of sub-populations. Therefore, absolute fluorescence levels should not be compared across different experiments. However, special care was always taken to ensure that the parameters affecting the PMT signal were kept constant during the acquisition of a given emulsion. In the specific case of Figure 2f, the PMT gain was increased during the acquisition of the blank emulsion, hence the higher fluorescence levels compared to the other panels. We have noted this in the revised version of the manuscript to avoid confusion.

3. Fig. 2c, the negative peak is at around 3.8, which is similar to the positive peak in Fig. 2a. Here the authors restrict the fitting of data into two peaks. However, as can be seen in Fig. 2a, a shoulder with small bumps at higher intensity is seen. If we assume some droplets contained a lot of bacteria, more peaks can be fitted. Then back to Fig. 2c, is it possible the negative peak would actually be a positive peak, and there is few droplet with 0 phages in? For an unknown sample without knowing the phage concentration, how would you exclude this possibility?

REPLY: We assume that negative droplets always exhibit a single-mode distribution, as shown in Fig. 2e–f, while positive droplets can display multiple modes due to successive lysis cycles, depending on the number of bacteria and the time elapsed after the first lysis. However, in our experience, positive peaks consistently show a characteristic shape that allows their identification. Notably, the additional positive peaks are always more right-skewed and flattened than the main positive peak. This distinctive distribution most likely results from the dynamics of additional lysis events occurring within droplets, which we did not investigate in detail.

If an experimenter encounters a complex signal distribution, a convenient approach would be to vary the mixing fraction. The evolution of the different modes with changes in the mixing fraction would clarify their identity and help avoid misinterpretation. Given how easily the mixing fraction can be adjusted during emulsification, this procedure can be applied systematically to ensure correct peak identification. We explicitly suggest this in the revised text.

4. Still in Fig. 2, Fig. 2b and Fig. 2d were to obtain the fraction of positive droplets in relation with phage titer based on Eq. 1, which depends on the of bacteria concentration (λ_b) and phage concentration (λ_p). However, based on the later sections, the authors found that not all bacteria is lysed by phages (“Screening with discrete mixing factions” and “screening with continuous mixing fractions”). That only a percentile of bacteria are infected with phages, and the percentile depends on the bacteria concentration, as shown in Fig. 3e. If so, what λ_b was applied in the analysis of Fig. 2b and Fig. 2d? How to rationale what number to apply when another bacteria is used?

REPLY: Indeed, our conclusion in Fig. 3 supports that λ_b critically influences either the kinetics of the lysis or the effective cell occupancy (cells that contribute to the lysis reaction). This phenomenon,

while hinting at some interesting lysis reaction dynamics does not impact the phage enumeration. As long as the cell encapsulation is not limiting the lysis reaction, equation 1 will provide the phage titer with the same accuracy. In other word, it is not required to know λ_b precisely as long as all droplets contain at least one bacterial cell. In order to achieve this, the experimenter just needs to set the OD to a sufficiently high value. In Fig 2. λ_b was deduced from the OD as in Figure 3.

5. On page 8, first paragraph, the authors attribute the discrepancy of obtained alpha to the designed alpha to that “phage adsorption occurs more slowly at lower bacteria densities”. If it was because of the slow adsorption rate, would this issue to be solved by longer culture time? As shown in Fig. 5, the fraction plateau after 1000 minutes, which is around 16 hours. Would a 16-hour culture solve the problem?

REPLY: Yes, indeed, incubating the emulsion for a longer period (as described in Figure 5, approximately 16 h) would likely allow lysis to complete even in droplets with different mixing fractions. Our focus was to highlight a non-intuitive linear trend, as observed in Figure 3, related to phage lysis dynamics. However, if the experimenter's goal is solely phage enumeration, a longer incubation time can be combined with a high bacterial density to accelerate lysis completion, consistent with the effect described above.

6. On page 9, paragraph 2, line 7, “the effective bacterial density determined via the fit was found to correspond to only 16.6% of that calculated from an OD measurement in bulk.” Only 17% of bacteria is in active state seems quite low. Is there any literature or other experiment to support this low number?

REPLY: The fit shown in Figure 4 allows us to extract an effective bacterial density. However, to compare this value to the total number of bacterial cells present, we rely on optical density (OD) measurements. The relationship between the number of cells per unit of volume and OD units depends on several parameters, such as the strain, the medium and the growth stage. So far, we have used the approximation 1 OD unit = 8×10^8 cells/mL. To address the reviewer's comment, we decided to rely on a value more representative of our specific conditions, determined experimentally. Therefore, we performed CFU-OD measurements using the same strain and medium and found a value of 4.17×10^8 cells/mL per OD unit. This lower value led to a smaller estimate of the total number of cells and increased the effective bacterial proportion to 28.6% instead of 16.6%. We updated all estimated bacterial density values based on OD measurements as well as computed values depending on it, in the main text, in the Supporting Information and in the data depository.

As for the proportion of host cells involved in lysis, the updated value of 28.6% still remains quite low. However, we found clear evidence in the literature regarding the influence of host physiological state on the dynamics of phage infection that could explain this low proportion. As mentioned in the manuscript, host cells with a smaller growth rate – such as in our case due to a non-supporting growth medium - cause a significant reduction in phage development¹⁻⁴. In some extreme cases, when host cells enter a true dormancy state (known as Type I persisters), lambda-mediated host phage lysis seems to occur only when the dormancy state is exited⁵. An even more compelling example, in our opinion, was provided by a study with lambda phage⁶ in which the authors demonstrated a dramatic protective effect of carbon starvation in host cells. After a 14-hour culture, a four-order-of-magnitude increase in the number of surviving host cells was observed due to starvation. Given these examples, the influence of the host cell's metabolic state on infectivity makes the observed low proportion of

lysed cells quite plausible. However, we cannot determine, among the remaining 71.4% of “non-active” cells, which fraction truly escaped infection and which were infected but exhibited delayed or incomplete phage development – thus remaining undetectable at the time of droplet acquisition. We expanded the discussion in the text on this point and also added the references.

1. Hadas, H., Einav, M. & Zaritsky, A. Bacteriophage T4 Development Depends on the Physiology of its host E. coli. *Microbiology (N Y)* 143, 179–185 (1994).
2. Golec, P., Karczewska-Golec, J., Loś, M. & Wegrzyn, G. Bacteriophage T4 can produce progeny virions in extremely slowly growing Escherichia coli host: Comparison of a mathematical model with the experimental data. *FEMS Microbiol Lett* 351, 156–161 (2014).
3. You, L., Suthers, P. F. & Yin, J. Effects of Escherichia coli physiology on growth of phage T7 in vivo and in silico. *J Bacteriol* 184, 1888–1894 (2002).
4. Nabergoj, D., Modic, P. & Podgornik, A. Effect of bacterial growth rate on bacteriophage population growth rate. *Microbiologyopen* 7, 1–10 (2018).
5. Pearl, S., Gabay, C., Kishony, R., Oppenheim, A. & Balaban, N. Q. Nongenetic individuality in the host-phage interaction. *PLoS Biol* 6, 0957–0964 (2008).
6. Łoś, M. *et al.* Effective inhibition of lytic development of bacteriophages λ , P1 and T4 by starvation of their host, Escherichia coli. *BMC Biotechnol* 7, 1–6 (2007).

Reviewer #3 (Remarks to the Author):

The manuscript of Givélet et al describes the application of droplet microfluidics to monitor phage-host bacteria interactions. The approach allows the quantitation of individual phage infection events. Overall the manuscript is well written and opens new possibilities for the phage host interaction studies.

I don't have any critical comments and noticed just a couple of typos in the text or references. Page 13, line 4 from bottom. Hammatsu written with 3 m's
Ref 6, the last author should be Harald Brüssow

Reviewer #3 (Remarks on code availability):

I am not competent to evaluate the code

REPLY: We would like to thank the reviewer for their positive assessment of our work. We have corrected the typos noted in the revised version of the manuscript.

Reviewer #1 (Remarks to the Author):

While the authors have clarified the number of replicates and the titer calculation formula, their response regarding the original DLA data remains unsatisfactory. The claim that the lack of an automated plaque counter prevented the capture of any original images is unconvincing, as standard laboratory practice involves photographing plates for documentation, regardless of the counting method. Providing representative images from different experiments is not an acceptable substitute for the primary data generated in this study. Furthermore, the reported 4-hour incubation period for visible plaque formation is highly atypical and raises concerns about the validity of the DLA results used for comparison. Therefore, we must insist that the authors provide the original raw data for the DLA assays. Specifically, how were the results recorded on the day of the experiment? A laboratory notebook entry or data sheet detailing the dilutions made, replicates, and incubation start time. The original plate count records. This is typically a table with columns for: Dilution Factor, Plate Identifier (e.g., Replicate A, B), Plaque Count, and Notes. Without this primary data, the foundation for comparing and validating their new method is significantly weakened.

Reply: We apologize for not being able to provide the specific pictures suggested by the reviewer. Such images are not generated in our experimental workflow, in which primary DLA data are documented through plaque counts are recorded in form of digital notes, sometimes using paper-based intermediate notes. The data is then transferred and consolidated into digital records (i.e. excel tables), where accurate titer calculations are performed.

This has been standard methodology in our lab for many years and we felt that reporting only titer values without further source data was common practice in phage literature. We found that even in very recent publications, source data for phage titers is rarely reported. E.g. the following:

- Xing, B., Liu, C., Chen, W. et al. Gut Phage Biobank: a collection of bacteriophages targeting human commensal bacteria. *Nat Commun* 16, 11050 (2025). <https://doi.org/10.1038/s41467-025-61946-0>
- Hu, H., Popp, P.F., Hughes, T.C.D. et al. Structure and mechanism of the Zorya anti-phage defence system. *Nature* 639, 1093–1101 (2025). <https://doi.org/10.1038/s41586-024-08493-8>
- Almashtoub, S.A.; Fares, G.H.; Abdo Ahmad, T.A.; Barada, S.; Turk, A.; Shoukair, D.; Matar, G.M.; Saba, E.S. Isolation and Characterization of a Novel Thermostable Bacteriophage Targeting Multi-Drug-Resistant *Salmonella* Enteritidis. *Viruses* 2025, 17, 1518. <https://doi.org/10.3390/v17111518>

However, we agree with the reviewer that it should be common practice to record pictures of DLA plates as part of a fully transparent experimental process and will do so in the future.

For this work, we provide the primary records used for DLA titer determination in the form of a comprehensive table reporting all DLA calculations, including the exact plaque counts used to compute each mean and standard deviation for every reported titer. We will make the table available alongside this reply, as well as in the data repository (<https://zenodo.org/records/18077663> new version v5), in an effort towards full source data transparency. In addition, with this reply, we provide a document, showing

exemplary photographs of original notes about plaque counts from the days on which the DLA experiments were performed. For a better clarity regarding DLA data, we also updated the Table 4 in Supplementary Information for a better readability.

The reviewer's comment on the 4 h incubation period of the DLA led us to realize that the wording used in the respective methods section, is misleading. The standard procedure is as follows: DLA plates were incubated and regularly observed. Plaques were visible from around 2,5 h, but evaluation (i.e. counting of plaques) was performed no earlier than after 4 h total incubation time, to ensure that plaques were fully developed and accurate plaque counts were obtained. This practice is in good agreement with the results obtained in the DLA time-series (Fig. 5e). We amended the methods section accordingly. While consolidating our DLA data, we identified minor miscalculations in some DLA standard deviation values. These values have been corrected in the revised Supplementary Information and in the data repository. We thank the reviewer for drawing attention to this point, which allowed us to further improve the accuracy and traceability of the reported data.

We appreciate the authors' clarification on the principle of extending the dynamic range by varying the mixing fraction (α). However, the response does not fully address the specific question raised. The question specifically asked for the real dynamic range for $\alpha = 0.260$, 0.353 , and 0.608 , assuming 10^6 droplets. The reply provides a plausible overall range (1.2×10^6 to 2.0×10^8 PFU/mL) achieved by varying α , but it fails to specify the effective range for each of the three individual α values used in Figure 3. To properly evaluate the method's performance and the data presented in Figure 3, it is essential to know the accurate dynamic range for each specific condition tested. The 5% to 95% positive droplet threshold is a reasonable criterion. Please provide a calculation or estimation of the upper and lower detection limits for each specific α value (0.260 , 0.353 , 0.608) based on this criterion and the stated total droplet number. This quantitative information is crucial for assessing the practical utility and precision of the technique for a given setup.

Reply: We apologize for the incomplete previous response, in which we reported only the outer bounds of the expanded dynamic range, corresponding to the lower bound for $\alpha = 0.764$ and the upper bound for $\alpha = 0.260$.

Using the same 5%/95% criterion and assuming 10^6 droplets, the dynamic ranges are ($3.5 \times 10^6 - 2.5 \times 10^8$), ($2.6 \times 10^6 - 1.5 \times 10^8$), and ($1.5 \times 10^6 - 8.7 \times 10^7$) PFU mL⁻¹ for $\alpha = 0.260$, $\alpha = 0.353$, and $\alpha = 0.608$, respectively.

We now provide an updated version of Fig. S4a showing both the upper and lower detection limits for all α values used in Fig. 3. This figure will be included in the revised Supplementary Information. This revision addresses the reviewer's comment by explicitly reporting the full detection range for all α values, rather than only the two bounding limits reported in our previous response, thereby improving the clarity and completeness of the methodological description.

Date	Plaque count									Phage stock titer [PFU/mL] Titer=Plaque count*dilution factor/volume plated								Titer			
	#Replicates	replicate:	1	2	3	4	dilution factor	vol. plated (mL)	replicate:	1	2	3	4	5	MEAN	STD	Figure	Dilution before emulsion	MEAN	STD	
12/7/2023	5		31	23	18	26	27	1.0E+08	0.1		3.10E+10	2.30E+10	1.80E+10	2.60E+10	2.70E+10	2.50E+10	4.85E+09	Figure 2	100	2.50E+08	4.85E+07
1/9/2024	2		21	26				1.0E+08	0.1		2.10E+10	2.60E+10				2.35E+10	3.54E+09	Figure 4	200	1.18E+08	1.77E+07
1/29/2024	2		26	11				1.0E+08	0.1		2.60E+10	1.10E+10				1.85E+10	1.06E+10	Figure 3	200	9.25E+07	5.30E+07
2/13/2024	6	in 2 dilutions	191	187	188			1.0E+07	0.1		1.91E+10	1.87E+10	1.88E+10			mean and std of 2 dilutions:		Figure 5	200	9.55E+07	2.06E+07
			13	19	26			1.0E+08	0.1		1.30E+10	1.90E+10	2.60E+10			1.91E+10	4.13E+09				

7.12 Plaque assay 10^8

3x LB Top agar 0.75% on LB } BIOcultures in
 3x NZCYM 0.5% on NZCYM } respective media

~ 3 h: LB NZCYM

31 20 } $2.65 \cdot 10^{10}$

23 27

18 X (dga slipped)

$\frac{18}{2.4} \cdot 10^{10}$

both: similar plaque size, mixture of pin-point up to 0.1

→ clear circles, no Halo

⇒ $2.5 \cdot 10^{10}$

source data for DLA 7/12/2023, corresponding to data in Figure 2

Plaque Assay WS 23/01 & 20/01 10^{-8} 2x each
 09.01.24 21 + 26 → 235 33 + 15 → 7,4

source data for DLA 1/9/2024, corresponding to data in Figure 4

T7 Titer BSA stocks 29.01
 A: [redacted]
 B: fresh stock → labeled B-test + plate
 → dilute up to 10^{-8} → 500 μ l 10^{-8}
 => plate 2x each
 A [redacted] } 10^{-8} [redacted]
 B 20/11 } $\rightarrow 1,85 \cdot 10^{10}$

source data for DLA 1/29/2024, corresponding to data in Figure 3

Data to Figure 5e - DLA time series

time of evaluation	time (h)	Plaque count						V (mL):	Phage stock titer [PFU/mL] Titer=Plaque count*dilution factor/volume plated						Titer						
		dilution factor: 1,0E+07			1,0E+08			0,1	dilution factor: 1,0E+07			1,0E+08			Figure	Dilution before emulsion	MEAN	STD			
t(0) = 12:45		replicate:	1	2	3	1	2	3	replicate:	1	2	3	1	2	3	MEAN	STD				
16:30	2,5 3,75		166	175	167	8	8	5		1,66E+10	1,75E+10	1,67E+10	8,00E+09	8,00E+09	5,00E+09	7,00E+09	1,73E+09	Figure 5	200	3,50E+07	8,66E+06
18:00	5,25		183	186	186	13	18	23		1,83E+10	1,86E+10	1,86E+10	1,20E+10	1,60E+10	2,20E+10	1,68E+10	3,20E+09			8,40E+07	1,60E+07
20:00	7,25		191	187	188	13	18	24		1,91E+10	1,87E+10	1,88E+10	1,30E+10	1,80E+10	2,30E+10	1,83E+10	3,18E+09			9,13E+07	1,59E+07
over night	20					13	19	26					1,30E+10	1,90E+10	2,60E+10	1,93E+10	6,51E+09			9,30E+07	1,75E+07
																				9,67E+07	3,25E+07

13.02 T7 Titer (BSI-Stock → X)

1/10⁻⁸

Time	1	2	3	1	2	3
2,5 h				8	8	5
3:45 (16:30)	165 (14)	175 (13)	167 (17)	12 (1)	16 (2)	22 (3)
18:00	+17 (17)	+11 (7)	+19 (16)	13	18	23
20:00	+8	+1	+2	-	-	+1
ON				13	19	26

→ 1,93 · 10¹⁰

+0 = 19:45

source data for DLA time series 2/13/2024, corresponding to data in Figure 5